# Real-Time Human Motion Tracking by Tello EDU Drone

**DOI:** 10.3390/s23020897

**Published:** 2023-01-12

**Authors:** Anuparp Boonsongsrikul, Jirapon Eamsaard

**Affiliations:** Department of Electrical Engineering, Faculty of Engineering, Burapha University Chonburi Campus, Chonburi 20131, Thailand

**Keywords:** motion tracking, drone, recognition

## Abstract

Human movement tracking is useful in a variety of areas, such as search-and-rescue activities. CCTV and IP cameras are popular as front-end sensors for tracking human motion; however, they are stationary and have limited applicability in hard-to-reach places, such as those where disasters have occurred. Using a drone to discover a person is challenging and requires an innovative approach. In this paper, we aim to present the design and implementation of a human motion tracking method using a Tello EDU drone. The design methodology is carried out in four steps: (1) control panel design; (2) human motion tracking algorithm; (3) notification systems; and (4) communication and distance extension. Intensive experimental results show that the drone implemented by the proposed algorithm performs well in tracking a human at a distance of 2–10 m moving at a speed of 2 m/s. In an experimental field of the size $95\times 35\;m^{2}$, the drone tracked human motion throughout a whole day, with the best tracking results observed in the morning. The drone was controlled from a laptop using a Wi-Fi router with a maximum horizontal tracking distance of 84.30 m and maximum vertical distance of 13.40 m. The experiment showed an accuracy rate for human movement detection between 96.67 and 100%.

## 1. Introduction

Tracking human movements is a challenging task applicable to a variety of fields, such as real-time exercise tracking [1,2,3,4,5], fall detection for the elderly [6,7,8,9,10,11], intrusion detection-based surveillance [12,13] in restricted areas such as railroad tracks and railway stations [14], etc. Wearable sensors [15,16,17,18] are popular for monitoring the health status in certain patients; however, these sensors only track certain vital signs and are insufficient for monitoring movement. Closed-Circuit Television (CCTV) and Internet Protocol (IP) cameras, on the other hand, can fully track movement, but are stationary and can seldom monitor remote areas, such as those where disasters have occurred.

Instead of CCTV and IP cameras, drones provide an alternative for tracking human motion, as they are mobile and can access almost any location quickly. Semi-autonomous unmanned aerial vehicles (UAVs) [19] are often used in emergency response scenarios for fire surveillance and search-and-rescue operations, in order to find people who are in distress or imminent danger. Nevertheless, automatic motion tracking remains a challenge. Researchers [20] have introduced a drone-based system for 3D human motion capture, in which the drone orbits a human subject, records a video, and reconstructs 3D full-body poses. The system uses an autonomously flying drone (DJI Phantom 4 or DJI Mavic Pro) with an on-board RGB camera. The DJI Mavic Pro comes with the active tracking feature and provides a bounding box of the tracked subject. However, in this work, we do not focus on a methodology for automatic human motion tracking; instead, we aim to recover the 3D pose sequence of the subject.

In contrast, our paper aims to present the design and implementation of human motion tracking by a Tello EDU drone, which does not come with an active tracking feature. To add the active tracking feature, we propose a human motion tracking algorithm implemented on a laptop, in order to exchange commands and responses to automatically control directions of the Tello EDU drone as it tracks people’s movements. Real-time streaming videos from the Tello EDU drone to the laptop controller are processed using machine learning. Text commands from the laptop controller are sent to the Tello EDU drone, in order to manage the flight plan. The communication frequency between the Tello EDU drone and the personal computer is 2.4 GHz, which is an unlicensed spectrum, the same as WiFi.

This research study is divided into four parts: First, a novel human motion tracking system for the Tello EDU drone is proposed. Second, a human motion tracking algorithm equipped with Google’s MediaPipe framework is presented. Third, a comprehensive evaluation of the proposed algorithm’s performance in human motion detection is provided. Finally, we compare our work with other current research and discuss future challenges and opportunities. Our work makes two contributions to current research: first, the proposed human motion tracking algorithm can enhance common drones, transforming them into smart drones that can automatically track human motion. Second, the proposed system of human motion tracking can be applied for search-and-rescue activities.

The remainder of this paper is organized as follows. Section 2 explains the hardware and software relevant to the Tello EDU drone (Section 2.1), MediaPipe framework (Section 2.2), and design and implementation of human motion tracking (Section 2.3). Section 3 describes the proper distances between a person and a Tello EDU drone (Section 3.1), the target’s speed and direction for human motion tracking by a Tello EDU drone (Section 3.2), the exploration of light intensity for human motion detection (Section 3.3), the notification system for human motion tracking (Section 3.4), fault detection (Section 3.5), and the comparison of flight-control distances between the Tello App and our design (Section 3.6). Section 4 reviews and compares our work and related work, as well as discussing future challenges and opportunities. Finally, Section 5 presents our conclusions.

## 2. Materials and Methods

This section explains the hardware and software relevant to human motion tracking using the Tello EDU drone. In terms of hardware, the Tello EDU drone is the main vehicle, controlled by a laptop computer using Wi-Fi. For software, the MediaPipe framework, imported and implemented in the Python language, is a tool that can be used for human gesture recognition. To design and implement our human motion tracking system, we first calculated 3D forms in the *X*, *Y*, and *Z* dimensions, where *X* is a coordinate representing a left or right target, compared to a screen center dot; *Y* is a coordinate representing an up or down target compared to a screen center dot; and *Z* is a coordinate indicating a big or small target size on the screen.

### 2.1. Tello EDU Drone

The Tello EDU drone is a small drone manufactured by Ryze Tech in cooperation with Scratch and Swift, as illustrated in Figure 1. Its design and size are intended for students or others interested in UAV technology. In addition, Ryze Tech has enabled Scratch and Swift to develop the program through the Python SDK, making it relatively easy to build functions and enhance the capabilities of the Tello EDU drone [21]. The drone can be controlled by a smartphone or a laptop computer using 2.4 GHz communication.

### 2.2. MediaPipe Framework

Human pose estimation from streaming video plays a vital role in many applications, such as quantifying physical exercises and sports. For example, it can evaluate the performance of athletes by counting the number of push-ups or squats; that of boxers by counting the number of jabs, hooks, or uppercuts; and so on. It can also enable the overlay of digital content and information on top of the physical world in augmented reality. MediaPipe Pose is a machine learning solution for body pose tracking, inferring all landmarks and a background segmentation mask for the whole body from RGB video frames. In MediaPipe Pose, the total number of human poses includes 33 landmarks [22,23], as illustrated in Figure 2.

The MediaPipe holistic framework [24] simultaneously tracks human pose, face landmarks, and hands in real-time, and can be divided into four steps. First, MediaPipe Pose is applied for estimation of the human pose and the subsequent landmark model. Second, the inferred pose landmarks are selected in three regions of interest, by cropping each hand (2×) and the face. Third, these three regions are employed in a re-crop model for improvement, and then are cropped with the full-resolution input frame. Task-specific face and hand models are applied to estimate their corresponding landmarks. Finally, all landmarks, comprised of 33 pose landmarks, 468 face landmarks, and 21 hand landmarks per hand, are merged, resulting in the full 540 landmarks being obtained.

### 2.3. Design and Implementation of Human Motion Tracking

In terms of the design and implementation of human motion tracking, the proposed method can be divided into four steps: (1) computer-based flight control; (2) human-tracking flight with the MediaPipe framework; (3) automatic photo capture and alarm systems; and (4) communication protocol and extended drone control ranges. An overview of human motion tracking by the Tello EDU drone is illustrated in Figure 3. To control the Tello EDU drone’s flight, a computer sends commands to a Tello EDU drone using a Wi-Fi router, while the Tello EDU drone provides responses to the computer. Our proposed algorithm performs human motion tracking. When the drone detects a person within its visibility radius and its detection time is greater than a certain threshold, the computer automatically captures pictures of the person.

#### 2.3.1. Computer-Based Flight Control

For the computer-based flight control, a graphic user interface (GUI) was designed using tkinter-python. This GUI is a control panel for controlling takeoffs, landings, directions, and speeds, as illustrated in Figure 4. This GUI also provides manual and automatic modes for the drone’s flight. No. 1 indicates the takeoff button. No. 2 indicates the button for moving forward. No. 3 indicates the button for moving left. No. 4 indicates the button for moving right. No. 5 indicates the button for moving backward. No. 6 indicates the button for moving up. No. 7 indicates the button for turning left. No. 8 indicates the button for turning right. No. 9 indicates the button for moving down. No. 10 indicates the landing button. No. 11–13 are various speed buttons. No. 14 indicates manual control, requiring that buttons No. 1–10 and 11–13 be used together. No. 15 denotes automatic control, requiring the MediaPipe framework and our proposed algorithm for human motion tracking. No. 16 terminates the GUI and system.

#### 2.3.2. Human-Tracking Flight using the MediaPipe Framework

The principle of tracking flight involves the use of overlapping coordinate vectors. In Figure 5, there are two circle points (red and green). The red circle point represents the center coordinates of the screen, which is a benchmark for comparison. The green circle point represents a target coordinate, located on an edge of a bounding box. When the two circles are apart, this means that the drone and the detected target are not properly aligned. To perform this alignment, the human body has to move to the center of the drone’s camera, or the drone’s camera has to move to the center of the human target.

In our coordinate system, *x* and *y* values are compared to the frame width and height, respectively. A magnitude of *z* represents the target depth, with the depth at the midpoint of hips being the origin. The documentation of MediaPipe Pose [22] states that the magnitude of *z* uses roughly the same scale as *x*. Therefore, our proposed algorithm adopted this concept for MediaPipe Pose [22]. Note that the target depth and target size are highly relevant; therefore, they can be considered as interchangeable in our explanation.

In our design and computation, the 3D target coordinate is written using a vector $\left[x_{i},y_{i},z_{i}\right]$, where $x_{i}=x_{min},y_{i}=y_{min}+150,z_{i}=x_{min}$. The number 150 is our design offset, shifted from the lower left corner of the bounding box. In our design, through trial and error by changing various offsets in vertical distances, the drone tended to crash into the ground if the offset was less than 150 and the person moved down onto the ground. Therefore, an offset of 150 was selected, in order to protect the drone from damage in all experiments.

Note that the $x_{min}$ and $y_{min}$ are values on the bounding box, which are updated every frame. The index *i* denotes the video frame, where 1<i<n and *n* is the last video frame of human motion tracking. To maintain human motion tracking, a 3D benchmark is compared to the target. The 3D benchmark coordinate determines the middle of a drone’s screen, presented as a vector $\left[x_{0},y_{0},z_{0}\right]$, where $x_{0}=w/2,y_{0}=h/2$, and $z_{0}=c$. The variables w,h, and *c* denote the width, height, and depth, respectively. In our design, *w* is 960, *h* is 720, and *c* is 456; thus, the 2D resolution is 960 × 720 and the target size is 456. The target vector and benchmark vector are designated by green and red dots, respectively, in Figure 6.

For a target size of $z_{0}$, a certain value is selected to maintain a safe distance between the target and the Tello EDU drone. This adopted number is taken from the set [1026, 684, 456, 304, 202, 136, 90], presented for the detection of face sizes by Jabrils [25].

In our design, through trial and error by changing various $z_{0}$ values, if $z_{0}$ is 1026 or 684, the target sizes were quite big and the measured distances were too close between the person and the Tello EDU drone. These thresholds resulted in crashing into the person when the person moved forward and abruptly stopped. On the other hand, with $z_{0}$ values between 90 and 304, the target sizes were small and measured distances were far. These numbers can cause the Tello EDU drone to lose track of human movements. Therefore, $z_{0}$ of 456 was selected, in order to stay safe enough to prevent the person from being injured while maintaining the drone’s tracking in all experiments.

The distance between the benchmark vector $\left[x_{0},y_{0},z_{0}\right]$ and a target vector $\left[x_{i},y_{i},z_{i}\right]$ is denoted as the distance vector $\left[\Delta x_{i},\Delta y_{i},\Delta z_{i}\right]$. This distance vector is used to control the movements of the Tello EDU drone, as summarized in Table 1. Due to the use of streaming video frames, these vectors become data series and their relation in our algorithm can be written using a matrix Equation (Equation 1), where B is a benchmark matrix (Equation 2), T is a target matrix (Equation 3), and D is a distance matrix (Equation 4):(1)B−T=D,
(2)$$B=\left[\begin{array}{ccc} x_{0} & y_{0} & z_{0}\\ x_{0} & y_{0} & z_{0}\\ \vdots & \vdots & \vdots \\ x_{0} & y_{0} & z_{0}\\ \vdots & \vdots & \vdots \\ x_{0} & y_{0} & z_{0} \end{array}\right],$$
(3)$$T=\left[\begin{array}{ccc} x_{1} & y_{1} & z_{1}\\ x_{2} & y_{2} & z_{2}\\ \vdots & \vdots & \vdots \\ x_{i} & y_{i} & z_{i}\\ \vdots & \vdots & \vdots \\ x_{n} & y_{n} & z_{n} \end{array}\right],$$
(4)$$D=\left[\begin{array}{ccc} \Delta x_{1} & \Delta y_{1} & \Delta z_{1}\\ \Delta x_{2} & \Delta y_{2} & \Delta z_{2}\\ \vdots & \vdots & \vdots \\ \Delta x_{i} & \Delta y_{i} & \Delta z_{i}\\ \vdots & \vdots & \vdots \\ \Delta x_{n} & \Delta y_{n} & \Delta z_{n} \end{array}\right].$$

Table 1 describes the distance conditions and drone movements in three dimensions. In the horizontal dimension, there are three conditions: when $\Delta x_{i}$ is greater than 100 (No. 1), $\Delta x_{i}$ is less than −100 (No. 2), and $\Delta x_{i}$ is between −100 and 100 (No. 3), the drone will turn left, turn right, and stay with no change (i.e., keeping the same movement as in the previous state), respectively. In the vertical dimension, there are also three conditions: when $\Delta y_{i}$ is less than −55 (No. 4), greater than 55 (No. 5), and between −55 and 55 (No. 6), the drone will move up, move down, and stay with no change, respectively. In the third dimension, there are two conditions: when $\Delta z_{i}$ is greater than zero (No. 7) and less than zero (No. 8), the drone will move backward and move forward, respectively.

Algorithm 1 describes the performed computation. When pose landmarks and consecutive frames are streamed, target coordinates on the bounding box are computed to find two minimum values, consisting of $x_{j}$ compared to $x_{j+1}$ and $y_{j}$ compared to $y_{j+1}$. In each iteration, $x_{i}$ is assigned by $x_{min},y_{i}$ is assigned by $y_{min}+150$, and $z_{i}$ is assigned by $x_{min}$. The target vector $x_{i},y_{i},z_{i}$ is compared to the benchmark vector $x_{0},y_{0},z_{0}$, resulting in the distance vector $\Delta x_{i},\Delta y_{i},\Delta z_{i}$. The values of the distance vector will determined the movements performed, with respect to the conditions (1–8) described in Table 1. Scenarios of movements are listed in the following.
**Algorithm 1** Human motion tracking algorithm.**Input:** Stream of pose landmarks
**Output:** Distance vector = $\left[\Delta x_{i},\Delta y_{i},\Delta z_{i}\right]$
 initialize $\Delta x_{i}=0$, $\Delta y_{i}=0$, $\Delta z_{i}=0$,
 $x_{i}=0$, $y_{i}=0$, $z_{i}=0$,   i=1, j=1,
 $x_{b_{j}}$=0, $y_{b_{j}}$=0,
 $x_{0}=\frac{w}{2}$, $y_{0}=\frac{h}{2}$, $z_{0}=TSize$
 **while** there are pose landmarks **do**
  update a target vector from two consecutive incoming frames by
  $x_{min}\leftarrow {\mathrm{argmin}}_{x}\left(x_{b_{j}},x_{b_{j+1}}\right)$;    ▹$x_{b_{j}}$ = $x_{i}$ on a boundary box
  $y_{min}\leftarrow {\mathrm{argmin}}_{y}\left(y_{b_{j}},y_{b_{j+1}}\right)$;    ▹$y_{b_{j}}$ = $y_{i}$ on a boundary box without a 150 offset
  $x_{i}\leftarrow x_{min};$
  $y_{i}\leftarrow \left(y_{min}+150\right)$;    ▹$y_{min}$ = $y_{i}$ on a boundary box plus a 150 offset
  $z_{i}\leftarrow x_{min};$
  **for** each target vector frame *i* **do**
   $\Delta x_{i}\leftarrow \left(x_{0}-x_{i}\right);$
   $\Delta y_{i}\leftarrow \left(y_{0}-y_{i}\right);$
   $\Delta z_{i}\leftarrow \left(z_{0}-z_{i}\right);$
   **if** $\Delta x_{i}$ > 100 **then**
    the drone turns left
   **else if** $\Delta x_{i}$ < −100 **then**
    the drone turns right
   **else**
    do the initialize action of $\Delta x_{i}=0$
   **end if**
   **if** $\Delta y_{i}$ < −55 **then**
    the drone moves up
   **else if** $\Delta y_{i}$ > 55 **then**
    the drone moves down
   **else**
    do the initialize action of $\Delta y_{i}=0$
   **end if**
   **if** $\Delta z_{i}$ > 0 **then**
    the drone moves backward
   **else if** $\Delta z_{i}$ < 0 **then**
    the drone moves forward
   **else**
    do the initialize action of $\Delta z_{i}=0$
   **end if**
  **end for**
  i←i+1;
  j←j+1;
 **end while**


Scenario of human motion tracking in horizontal direction

In our algorithm, the benchmark vector is fixed at $\left[480,y_{0},z_{0}\right]$. Suppose a target vector is $\left[300,y_{i},z_{i}\right]$, such that the target stays at the right-hand side. This results in a distance vector of $\left[180,\Delta y_{i},\Delta z_{i}\right]$ at a certain time (*T* = *t*). As $\Delta x_{i}$ is equal to 180, under condition No. 1 in Table 1 ($\Delta x_{i}>100$), the drone will turn left at a later time (T>t). Suppose the target vector is changed to be $\left[390,y_{i},z_{i}\right]$. Then, the distance vector will be $\left[90,\Delta y_{i},\Delta z_{i}\right]$. Then, there is no change, under condition No. 3 in Table 1, and the drone maintains the previous state of turning left. This scenario is illustrated in Figure 7.

In our algorithm, the benchmark vector is fixed at $\left[480,y_{0},z_{0}\right]$. Suppose a target vector is $\left[600,y_{i},z_{i}\right]$, such that the target stays at the left-hand side. This results in a distance vector of $\left[-120,\Delta y_{i},\Delta z_{i}\right]$ at a certain time (*T* = *t*). As $\Delta x_{i}$ is equals to −120, under condition No. 2 in Table 1 ($\Delta x_{i}<-100$), the drone will turn right at a later time (T>t). Suppose the target vector is changed to $\left[570,y_{i},z_{i}\right]$. Then, the distance vector will be $\left[-90,\Delta y_{i},\Delta z_{i}\right]$, and there will be no change, according to condition No. 3 in Table 1, and the drone keeps the previous state of turning right. This scenario is illustrated in Figure 8.

2.Scenario of human motion tracking in the vertical direction

In our algorithm, the benchmark vector is fixed at $\left[x_{0},360,z_{0}\right]$. Suppose a target vector is $\left[x_{i},460,z_{i}\right]$. Note that the number of 460 has been integrated, with an offset of 150. This results in a distance vector of $\left[\Delta x_{i},100,\Delta z_{i}\right]$ at a certain time (*T* = *t*). As $\Delta y_{i}$ is equal to −100, under condition No. 4 in Table 1 ($\Delta y_{i}<-55$), the drone will move up at a later time (T>t). Suppose the target vector is changed to be $\left[x_{i},320,z_{i}\right]$. Then, the distance vector is $\left[\Delta x_{i},40,\Delta z_{i}\right]$. There will be no change, according to condition No. 6 in Table 1, and the drone will maintain the previous state of moving up. This scenario is illustrated in Figure 9.

In our algorithm, the benchmark vector is fixed at $\left[x_{0},360,z_{0}\right]$. Suppose a target vector is $\left[x_{i},260,z_{i}\right]$. Note that the number of 260 has been integrated, with an offset of 150. This results in a distance vector of $\left[\Delta x_{i},-100,\Delta z_{i}\right]$ at a certain time (*T* = *t*). As $\Delta y_{i}$ is equal to 100, under condition No. 5 in Table 1 ($\Delta y_{i}>55$), the drone will move down at a later time (T>t). Suppose the target vector changes to $\left[x_{i},400,z_{i}\right]$. Then, the distance vector will be $\left[\Delta x_{i},-40,\Delta z_{i}\right]$, and there will be no change, under condition No. 6 in Table 1. Thus, the drone keeps the previous state of moving down. This scenario is illustrated in Figure 10.

3.Scenario of human motion tracking in backward and forward directions

In our algorithm, the benchmark vector is fixed at $\left[x_{0},y_{0},456\right]$. Suppose a target vector is $\left[x_{i},y_{i},446\right]$. This results in a distance of $\left[\Delta x_{i},\Delta y_{i},10\right]$ at a certain time (*T* = *t*). As $\Delta z_{i}$ is equal to 10, under condition No. 7 in Table 1 ($\Delta z_{i}>0$), the drone will move backward at a later time (T>t). Suppose the target vector is changed to be $\left[x_{i},y_{i},450\right]$. Then, the distance vector will be $\left[\Delta x_{i},\Delta y_{i},6\right]$. This is still under condition No. 7 in Table 1 and, so, the drone still moves backward. This scenario is illustrated in Figure 11.

In our algorithm, the benchmark vector is fixed at $\left[x_{0},y_{0},456\right]$. Suppose a target vector is $\left[x_{i},y_{i},466\right]$. This results in a distance of $\left[\Delta x_{i},\Delta y_{i},-10\right]$ at a certain time (*T* = *t*). As $\Delta z_{i}$ is equal to −10, under condition No. 8 in Table 1 ($\Delta z_{i}<0$), the drone will move forward at a later time (T>t). Suppose the target vector is changed to be $\left[x_{i},y_{i},460\right]$, then a distance vector will be $\left[\Delta x_{i},\Delta y_{i},-4\right]$. This is under condition No. 8 in Table 1, such that the drone still moves forward. This scenario is illustrated in Figure 12.

#### 2.3.3. Automatic Photo Capture and Alarm Systems

To verify that our algorithm is able to track human movement, when the Tello EDU drone has detected human movements for more than 5 s, the program is designed to capture 10 human pictures, provide an alarm stating “please check this picture and close this page to watch the real-time video,” and create a folder to store the series of 10 pictures. These actions are performed using the python functions cv2.imwrite(), engine.say(), and os.mkdir(), respectively. A time stamp is created using the function datetime.now().strftime(), in order for each detection time to have a different recorded name in the folder, providing convenience of verification.

#### 2.3.4. Communication Protocol and Extended Drone Control Ranges

In our design, the communication protocol between a computer and the Tello EDU drone is UDP. For commands provided by the libraries of Tello EDU drones, such as taking off, landing, streaming video on, streaming video off, moving up, moving down, turning left, turning right, moving forward, moving backward, flipping, etc., these commands are sent from a computer to the drone through UDP port No. 8889. On the other hand, responses and streaming video are sent back from Tello EDU drone to the computer through UDP port Nos. 11111 and 8890, respectively. Referring to Section 2.3.2 for details of the human-tracking flight using the MediaPipe framework, exchanging commands and responses is carried out during human motion tracking by the Tello EDU drone, according to our proposed algorithm. In the extended drone control ranges of our design, a D-LINK DIR-1251 Wi-Fi router plays an important role, as a repeater of 2.4 GHz signals. The aim of extended drone control ranges is to provide a larger area of human motion tracking. An overview of the communication protocol and extended drone control ranges is illustrated in Figure 4.

## 3. Outputs and Experimental Results

For design outputs, indicators are presented for convenience of verification, as shown in Figure 13. In the upper left-hand corner, the text “Detect” is a Boolean indicator of human detection. Then, “Time detect” refers to the duration of human detection. The coordinate $\left[\Delta x_{i},\Delta y_{i},\Delta z_{i}\right]$ shows a distance vector, such as [259,119,235], as the output part of Equation (Equation 4). In the upper right-hand corner, the date and time stamp of human detection are provided. Next, the battery usage is shown, in terms of percentage of remaining energy. In the lower left-hand corner, this text indicates the (automatic or manual) piloting mode of the Tello EDU drone.

All 33 pose landmarks are presented by connected lines and dots over the whole body, including the nose, mouth, left and right cheekbones, left and right shoulders, left and right arms, and so on. Both a bounding box (with four thicker corners) and target lock (without 4 thick corners) are presented in a red frame. At the left edge of the bounding box, the green circle represents the target coordinate, while the red circle of target lock represents the benchmark coordinate. Generally speaking, the closer the distance between the red and green circles, the higher the accuracy in human motion tracking. To present the human motion tracking performance in different aspects and environments, all experiments were divided into six groups. The first experiment presents proper distances between a human and the Tello EDU drone, with the experimental location being a drone sandbox. The second experiment investigated the target speed for human motion tracking by the Tello EDU drone, again using the drone sandbox. The third experiment explored the effect of light intensity on human motion tracking, with the experimental location being the outdoor parking lots of the Nanachart Bangsaen Hotel. For all experiments, a determined frame rate was 25 frames per second [25], and the *x*, *y* dimensions were 960 × 720. Visibility is a value from 0 to 1, indicating the likelihood of a landmark being visible in the image. A visibility of 1 (100%) is the best and that of 0 (0%) is the worst for human detection. Visibility is a real-time response that can be measured, as defined in Equation (Equation 5), where $z_{0}$ is a target benchmark of 456 and $z_{i}$ is the target size calculated by the MediaPipe framework.
(5)$$\mathrm{Visibility}=1-|\frac{z_{i}-z_{0}}{z_{0}}|.$$

The fourth experiment tested warning messages in our notification system, where the experimental location was outdoors. The fifth experiment was conducted to explain the distance extension for controlling the Tello EDU drone, carried out in the outdoor parking lot of Nanachart Bangsaen Hotel. Finally, the sixth experiment examined the fault detection performance, where the experimental location was outdoors, in an abandoned garden. Note that the terms “human” and “target” are considered interchangeable, as the target in this paper is a human.

### 3.1. Proper Distances between Person and Tello EDU Drone

In this experiment, we aimed to determine a range for human motion tracking based on our design and algorithm, where the experimental area was indoors. The range of 11 m was equally divided into 11 parts. The Tello EDU drone was kept at station No. 0, while a person stepped forward from station No. 11 (furthest from station No. 0) to station No. 1 (closest to station No. 0). The period of visibility measurement was 1 min at each station. For visibility measurements, traffic cones representing stations are illustrated in Figure 14a. An example of acceptable distances in human motion tracking is shown in Figure 14b.

For this experiment, a person repeatedly walked forward toward the Tello EDU drone, with 10 rounds. The experimental results in Figure 15 show that the benchmark of 456 implemented in our algorithm provided visibility between 78–95% when a target moved in the range between 2–10 m. In other words, the difference in size between the benchmark and the target was merely 5–22%. As the range between 2–10 m presented good visibility, this range was considered a good choice for determining other performance indices relevant to human motion tracking.

On the other hand, visibility was between 8–30% at 1 m. The main reason that relatively low visibility was obtained was that the MediaPipe framework prefers to detect the whole body, including the midpoint of the hips. Due to the short distance between the drone and the target, it is possible that only the upper half of the target was detected, as illustrated in Figure 11. At a distance of 11 m, the visibility was between 9–20%. In other words, the difference in size between the benchmark and the target was between 80–91%. As the distances of 1 m and 11 m presented poor visibility, they were not selected for the determination of other performance indices relevant to human motion tracking. Note that the dotted line represents a moving average graph from 1 to 11 m. We selected this moving average graph as the variation of visibility was low at each distance.

### 3.2. Target Speed and Direction for Human Motion Tracking by Tello EDU Drone

This experiment investigated the relationship between movement speeds and human motion tracking and visibility, where a person moves with different motion gestures. The human motion directions consisted of moving left, moving right, moving forward, and moving away from the Tello EDU drone. When moving forward and away from the Tello EDU drone, the motion gestures consisted of straight walking, straight running, zigzag walking, and zigzag running. When moving left and right through the Tello EDU drone’s camera, motion gestures consisted of left walking, right walking, left running, and right running. For this experiment, each motion gesture was repeatedly performed, for 10 rounds. In general, the walking speed of people ranges from 0.97–1.34 m/s [26]. Therefore, these target speeds and higher speeds were considered, in order to determine the maximum speed for this experiment.

For the straight walking test, the average speed of straight walking toward and that of straight walking away from the Tello EDU drone were 0.7776 and 0.7716 m/s, respectively. For the straight running test, the average speed of straight running toward and away from the Tello EDU drone were 1.6340 and 1.6234 m/s, respectively, as illustrated in Figure 16.

For the zigzag walking test, the average speed of zigzag walking toward away from the Tello EDU drone were 0.6536 and 0.6711 m/s, respectively. For the zigzag running test, the average speed of zigzag running toward and away from the Tello EDU drone were 1.5974 and 1.6181 m/s, respectively, as illustrated in Figure 17.

For straight walking through the Tello EDU drone’s camera, the average speed of straight walking from left to right and that from right to left were 1.1834 and 1.2019 m/s, respectively. For straight running through the Tello EDU drone’s camera, the average speed of straight running from left to right and that from right to left were 2.0661 and 2.1231 m/s, respectively, as illustrated in Figure 18.

As shown in Figure 19, visibility significantly decreased as human speed increased. The dotted line represents the polynomial trend-line of visibility, indicating the ability to track human motion gestures at different distances. In the trend line, visibility decreases by 10%, from 100% to 90%, when human motion speed increased from 0.67 to 1.38 m/s; meanwhile, visibility decreased 30%, from 90% to 60%, when human motion speed increased from 1.38 to 2.12 m/s. In each movement gesture speed, zigzag walking speed between 0.6536 and 0.6711 m/s provided visibility of 99% on average. At straight walking speed between 0.7716 and 0.7776 m/s, the Tello EDU drone had visibility of 99% on average. When going from left to right (and vice versa) through the Tello EDU drone’s camera at straight walking speed between 1.1834 and 1.2019 m/s, the Tello EDU drone had visibility of 97.5% on average.

At zigzag running speed between 1.5974 and 1.6181 m/s, a majority of visibility results were around 97%, but a few outliers were 0%. At straight running speed between 1.6234 and 1.634 m/s, the Tello EDU drone had a visibility of 96.5% on average. Running from left to right (and vice versa) through the Tello EDU drone’s camera at straight running speed between 2.0661 and 2.1231 m/s, the Tello EDU drone had uncertainty in human motion tracking, providing visibility of 60% on average.

From the experimental results, when the maximum target speed was 1.635 m/s, human motion tracking with a visibility of 96.5% was obtained. Comparing the experimental results between straight forward and transverse directions, the Tello EDU drone preferred the forward direction over the transverse direction for human motion tracking. Note that the dotted line represents a polynomial graph. We selected this trend-line as visibility becomes uncertain when increasing the target speed.

### 3.3. Exploration of Light Intensity for Human Motion Detection

The next experiment explored light intensity in an outdoor environment for human motion detection throughout a whole day. The experiment was divided into four time periods, consisting of the morning (9.00–10.00), noontime (12.00–13.00), afternoon (17.00–18.00), and evening (18.15–18.30). The Lux Light Meter Pro mobile application (version 2.2.1), installed on an iPhone, was used as an instrument to examine the light intensity. Visibility and light intensity are measured at each station, with a total of 10 stations. A total length of 10 m was equally divided 10 parts, and the Tello EDU drone stayed at the origin point at 0 m.

Figure 20 and Figure 21 show the results of experiments conducted in the morning (during 9.00–10.00) and at noontime (during 12.00–13.00), respectively. Visibility and light intensity were measured at every station, where a target stopped for a while and stepped forward from 1 to 10 m, towards the Tello EDU drone. The morning period provided a light intensity of 6773 lux on average, while the noon period provided a light intensity of 41,851 lux on average. Note that there was a difference in the target’s shadows between the morning and noontime periods. The target’s shadow in the morning stays at the back of the right hand, while the target’s shadow at noon stays in front of the right hand. Sunshine directions and target’s shadows have impacts on visibility; nevertheless, at a distance of 1 m, the Tello EDU drone was able to detect human motion, even at the visibility of 21.10% at noon.

Figure 22 and Figure 23 show the experiments conducted in the afternoon (during 17.00–18.00) and evening (during 18.00–18.15), respectively. The target stopped for measurement of light intensity at every station and stepped toward the Tello EDU drone from 1 to 10 m. Note that there were no shadows on the target in the afternoon. The afternoon period provided a light intensity of 519 lux on average. At a distance of 1 m, the Tello EDU drone was able to detect human motion, even at the visibility of 29% in the afternoon. The evening period provided a light intensity of 50 lux on average. At a distance of 1 m, the Tello EDU drone was unable to detect human motion, due to darkness, even with a visibility of 21.19% in the evening.

Figure 24 shows the experimental results over the whole day. By comparison of polynomial trend-lines, the visibility results were best in the morning, followed by in the afternoon, at noontime, and worst in the evening, respectively. The experimental results for each period are described as follows.

In the morning (during 9.00–10.00), the light intensity was 6773 lux on average. Visibility varied between 25–91% in the morning, providing the best result when compared to other periods. A target standing far away to 5 m from the Tello EDU drone resulted in better visibility (88% on average), compared to other distances, in the morning.

At noontime (during 12.00–13.00), the light intensity was 41,851 lux on average. Even though the light intensity was highest, the visibility was between 20–62%, making it the third best result among all periods. As the sunshine direction and target’s shadows have certain impacts, the visibility at noon was not as good as expected. A target standing far away to 4 m to the Tello EDU drone resulted in better visibility (60% on average), compared to other distances, at noontime.

In the afternoon (during 17.00–18.00), the light intensity was 519 lux on average. The visibility varied between 20–80% in the afternoon, providing the second-best result, compared to the other periods. A target standing far away to 4 m from the Tello EDU drone resulted in better visibility (75% in average), compared to other distances, in the afternoon.

In the evening (during 18.15–18.30), the light intensity was 50 lux on average. The visibility varied between 0–22% in the evening, giving the worst result, compared to other periods. Due to darkness in the evening, the Tello EDU drone was unable to detect human motion, even though there was 20% visibility on average at a distance of 1 m in the evening.

With gradual changes in light intensity and shadow conditions, altering an offset of 150 in the vertical direction is a feasible solution for visibility improvement, as the positional angle of the drone for human detection is changed with the offset. Adaptive changing the offset for visibility improvement is a challenge that should be addressed in further research.

### 3.4. Notification System of Human Motion Tracking

The next experiment aimed to test the verification of our notification system, as mentioned in Section 2.3.3. In the protocol, when the detection time of human motion is greater than 5 s, the program automatically takes 10 pictures, creates a folder to store them, and also states “please check this picture and close this page to watch the real time video.” Figure 25 shows a whole display, including a control panel on the right-hand side and the first picture saved in a created folder on the left-hand side. Figure 26 shows the experimental results when the Tello EDU drone took a series of 10 pictures when the detection period was higher than a threshold. The program also creates a folder to store the series of 10 pictures. This created folder is, for instance, named Human 2021-12-07_15-32-58, where 2021-12-07 is the date stamp and 15-32-58 is the time stamp. Note that the frame rate is 25 frames per second; therefore, 10 pictures can be simply stored within 1 min.

Furthermore, the alert system features a 5 s delay prior to sending human motion alerts, in order to avoid sending alerts under peripheral conditions where the background behind the target shows something like a tree with two branches or a gable roof, leading into the fault detection mentioned in Section 3.5. Based on our experiments, this fault detection occurs within less than 5 s, which is the main reason for using the 5 s delay prior to sending human motion alerts.

### 3.5. Fault Detection

Next, we examined fault detection in human motion tracking. The experimental field was an abandoned garden consisting of grasses, trees, and house roofs located next to walls, as illustrated in Figure 27. This area is selected as its background is slightly sensitive to human detection. For instance, it contains a tree with two branches which looks like a person raising their arms, as well as a gable roof that looks like a person’s head. These may lead to false positive and false negative results. In each experimental round, the distance between the drone and person ranged from 2 m at the beginning to 10 m at the end, where the person moved from left to right (and vice versa) through the Tello EDU drone’s camera.

The experimental results can be classified into four cases. First, when a person appears in front of the drone’s camera and the program correctly detects the person, this detection is a true positive, as illustrated in Figure 28. Second, when nobody is in front of the drone’s camera and it correctly detects nothing, this detection is a true negative, as illustrated in Figure 29. Third, when nobody is in front of the drone’s camera, but it incorrectly diagnoses that somebody (e.g., a tree having two branches) is in front of the drone’s camera, this fault detection is a false positive, as illustrated in Figure 30. Finally, when a person appears in front of the camera but it incorrectly diagnoses that something (e.g., gable roofs) is a target, instead of a person, this fault detection is a false negative, as illustrated in Figure 31.

In this experiment, there were 10 rounds of drone takeoffs, where each round included 300 s (5 min) for human motion tracking. As the frame rate was 25 frames per second, a total number of 7500 frames was considered in each round. The performance index used in this experiment was sensitivity [27] (also called detection rate or true positive rate [9]), which refers to ability to correctly track human motion, as defined in Equation (Equation 6).
(6)$$\mathrm{Sensitivity}=\frac{\mathrm{No}.\;\mathrm{of}\;\mathrm{true}\;\mathrm{positives}}{\mathrm{No}.\;\mathrm{of}\;\mathrm{true}\;\mathrm{positives}\;+\;\mathrm{No}.\;\mathrm{of}\;\mathrm{false}\;\mathrm{negatives}}.$$

The experimental results indicated that false negatives occurred four times—two times in the first round and one time in each of the second and eighth rounds. Each false negative occurred for less than 5 s and, so, there is no alarm for human detection. After a moment, the drone could return to human motion tracking. In the first round, the number of false negatives and that of true positives were 250 frames and 7250 frames, respectively, giving a sensitivity of 96.67%. In the second and eighth rounds, the number of false negatives and true positives were 125 frames and 7370 frames, with a sensitivity of 98.33%. In the rest of the experimental rounds, there were no false negatives, such that true positives were obtained for 100% of the 7500 frames. This correctness of detection is shown, in terms of sensitivity, in Figure 32.

For false positive detection, fault detection rarely occurred when nobody was in the frame in this environment. The drone detected trees or pillars instead of human motion. However, this false positive detection also lasted less than 5 s, quickly returning to true negative detection. All experiments showed that our design program equipped with the MediaPipe framework is efficient for human motion tracking, even in different areas.

### 3.6. Comparison of Flight-Control Distances between Tello App and Our Design

The next experiment aimed to compare the flight-control distances between the Tello App and our designed program with the Tello App. The experimental field was a rectangular outdoor area of size 95 × 35 m^2^, in a parking lot of the Bangsaen International Hotel next to Bangsaen beach in Chonburi province, Thailand. The Tello App functions strongly support real-time image transmission interface and camera video recording, while our design program provides similar functions to the Tello App, but also providing outstanding human motion tracking. To evaluate our program, comparison of flight-control distances between Tello App and our design was taken as a performance index assisting in the development of our design. In flight-control experiments, the drone-flight direction was from the west to the east. These flight-control distances can be divided into two cases: Normal and extended distance.

The first part of the experiment considered a normal flight-control distance without a Wi-Fi router. The drone directly communicated with a laptop controller. For horizontal distances, our program on a laptop and the Tello App on an iPhone provided the longest distances of 33 m and 35 m, respectively. For vertical distances, our program and the Tello App provided the highest distances of 7.3 m and 7.8 m, respectively. Second, we investigated extending the flight-control distances. A D-LINK DIR-1251 Wi-Fi router played an important role in serving as an intermediary communicator between the drone and the laptop controller. For horizontal distances, our program and the Tello App provided the longest distances of 84.3 m and 84.2 m, respectively; meanwhile, for vertical distances, our program and the Tello App provided the highest distances of 13.40 m and 30 m, respectively.

The comparison of flight-control distances between the Tello App and our design program is summarized in Table 2. In the majority of experimental results, Tello App provided longer flight-control distances than our design program. In particular, the vertical flight-control distance with a Wi-Fi router was more than two times greater. Our design program still provided drone fight in a vertical distance of 13.40 m, which is sufficient to obtain a wide view for detection of humans and, so, can be applied for human motion tracking, surveillance, and so on. For the rest of the flight-control distances, the difference between Tello App and our program was 2 m in the horizontal direction at maximum. Our design program facilitates drone flight with a horizontal distance of 33 m. This distance is sufficient for human motion tracking, surveillance, and so on. If a higher flight-control distance is required, a Wi-Fi router with stronger RSSI signal strength can be selected instead. Regarding the pros and cons, our design program is outstanding in human motion tracking, but presents certain limitations in flight-control distance, when compared to the Tello App.

### 3.7. Automatic Tracking of Human Motion

This section aims to show that our proposed algorithm is able to track human motion; in particular, we detail experiments of human motion tracking in two scenarios. First, the drone follows a person in an indoor–outdoor scenario. Second, the drone tracks a person moving in a multi-person scenario.

Drone following a person in an indoor–outdoor scenario

When following a person in an indoor–outdoor scenario, the drone must first take off and move using manual control, then applied for tracking human motion using automatic control. When the drone detects a person within its visibility radius, it keeps following the person on a path. A video of the view of the drone was recorded while a person randomly walked, as shown in Figure 33. The number 1 denotes the frame at the starting point, while 87 denotes the frame at the end point. The numbers 19, 47, 60, and 70 denote example frames, where the person moves toward the drone. A hundred sampling frames were selected from the recorded video of human motion tracking. To be concise, in this section, pictures A1–A12 were selected from frame No. 1–30, as illustrated in Figure 34; pictures A13–A24 were selected from frame No. 31–60, as illustrated in Figure 35; and Pictures A25–A36 were selected from frame No. 61–87, as illustrated in Figure 36. In Figure 36, it is shown that the drone can track a human that is randomly walking on the path from point No. 1 to point No. 87. In the hundred selected frames, different movement actions, consisting of moving forward, moving backward, turning left, turning right, flying up, and flying down, were separately counted. The total number of forward movements was 72, which was the highest number, thus corresponding to the most frequent movements of the drone. The total number of backward movements was 25, which occurred when a person moves forward or stops. The drone has to strike a balance between forwards and backwards, in order to maintain a good distance between the drone and the person. The total number of turning left motions was 27, while the total number of turning right motions was 68, corresponding to the more frequent turning right of the drone, which was following on a path from left to right, as illustrated in Figure 37.

2.Drone following a person in a multi-person scenario

When tracking a person in a multi-person scenario, our proposed algorithm is designed to detect the person closest to the drone. At first, in the experiment, a new person comes closer to the drone, compared to a first person tracked by the drone, following which the new person is tracked by the drone instead, as illustrated in B1–B12 of Figure 38.

For a second experiment, the distance between the new person and the drone is the same as the distance of the first person tracked by the drone. In B13–B16 of Figure 39, when a new person came to the first person and the first person moved away from the newcomer, then the drone still tracked the first person. When the first person came to the newcomer and quickly moved forward to stand behind the drone, the drone then tracked the newcomer instead, as illustrated in B17–B20 of Figure 39. When the first person stood and the newcomer walked past the first person being tracked by the drone, then the first person was still tracked by the drone, as illustrated in B21–B24 of Figure 39.

In both experiments, where a distance of the newcomer was closer to and the same as that of the first person, the experimental results showed that our algorithm detected the person closest to the drone, and kept tracking the first person if the newcomer stayed at the same distance as the first person.

## 4. Discussion

A previous study [28] has proposed 3D tracking of human motion using visual skeletons and stereoscopic vision, monitored by two static cameras in a single room. Another study [9] has proposed a fall detection scheme, monitored by a camera in a room. In [8], Raspberry Pi 2 equipped with a Cortex 900 MHz ARM Cortex^TM^-A7 processor was used, tested in a laboratory and a house, requiring six webcams for six rooms. However, these schemes [8,9,28] tested only indoor scenarios and did not provide details regarding lighting conditions, target movement speeds, and target distances away from a camera.

In [19], the authors designed a User Interface (UI) using an Angular application to support situational awareness in Cyber–physical Systems, in which humans and drones work together to support an emergency response. They used YOLOv3 as a pre-trained real-time object detection algorithm to support image recognition, and also provided visibility for detecting victims. Another study [20] has proposed a system for human motion capture using a drone, aiming to address the limitations of existing human motion-capture systems which rely on markers and static objects. However, one of selected drones in this work was a DJI Mavic Pro [29], which comes with the active tracking feature. However, the scheme of [19] required GPS features, while the cost of the DJI Mavic Pro used in [29] is higher.

In [30], the authors designed and implemented a complete fiducial marker based human head pose estimation system. However, the current system could only detect the rotation of an object’s head in a z-axis. In [31], the authors studied 2 methodologies in tracking human body movements by using the Tello EDU drone. The first methodology used the open pose library and gave very bad results in terms of response time by the drone. The second methodology adopted the detection of colors through the OpenCV library using 3 different color cards and positions. The biggest problem in this methodology was that during the day, the light varies and at some time during the day, the colors were not recognized. In [32], the authors presented a real time fall detection system with a back propagation neural network that could trigger an alarm people once a fall event occurs. However, the proposed scheme were stationary in monitoring.

In comparison with the previous schemes [8,9,19,20,28,30,31,32], our proposed scheme shows comprehensive performance in various experiments, considering light conditions, target movement speeds, and target distances, covering both indoor and outdoor scenarios. Furthermore, our scheme does not require extra hardware (e.g., GPS) or a drone that comes with the active tracking feature. Table 3 provides a comparison of human motion detection systems in the literature.

Thermal noise when flying a drone can cause the drone to become unstable. This thermal noise occurs when the drone has a continuous flight over a long time. In further research, we intend to develop a new algorithm for human motion tracking that reduces thermal noise while maintaining the visibility and sensitivity performance. In addition, we plan to enhance the feature that the drone can keep following the same person in multi-person scenarios, as well as studying different visibility angles of the drone in the human detection task.

## 5. Conclusions

In this paper, we presented the design and implementation of a human motion tracking program using a Tello EDU drone. The design methodology consisted of a control panel design, the human motion tracking algorithm, the notification system, and the communication and distance extension. The comprehensive experimental results demonstrated that the proposed algorithm performs well when tracking a human at a distance of 2 to 10 m, with the human movement speed limited at 2 m/s, and provides a sensitivity of human detection between 96.67 and 100%.

## Figures and Tables

**Figure 1 sensors-23-00897-f001:**
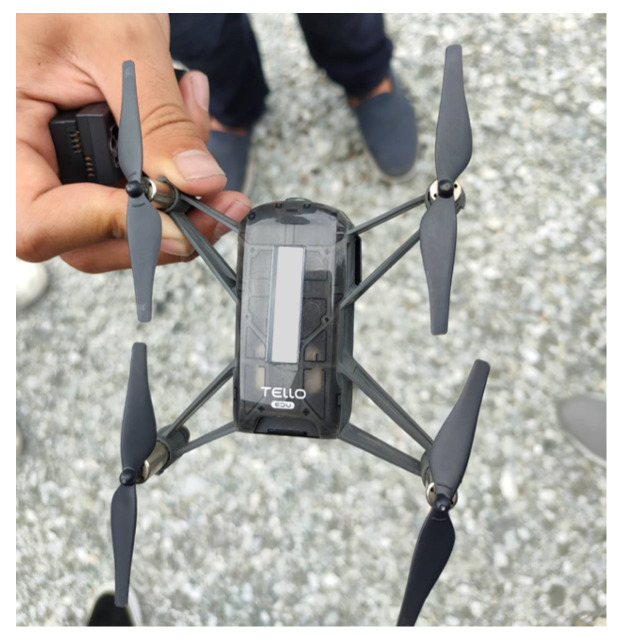
Tello EDUdrone.

**Figure 2 sensors-23-00897-f002:**
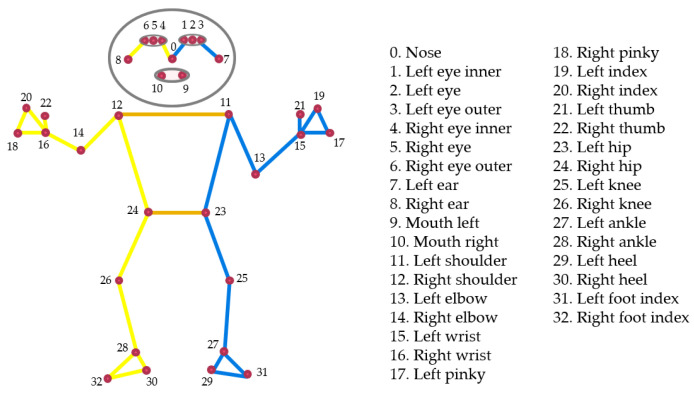
Pose landmarks.

**Figure 3 sensors-23-00897-f003:**
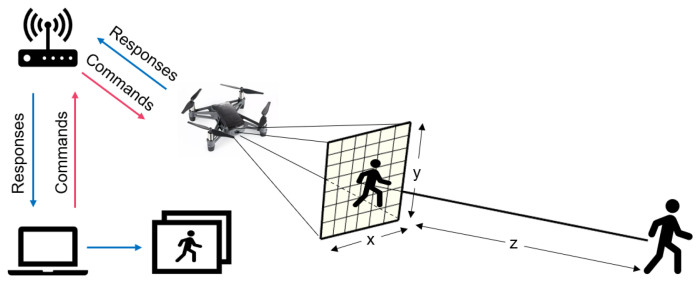
Overview of human motion tracking using Tello EDU drone.

**Figure 4 sensors-23-00897-f004:**
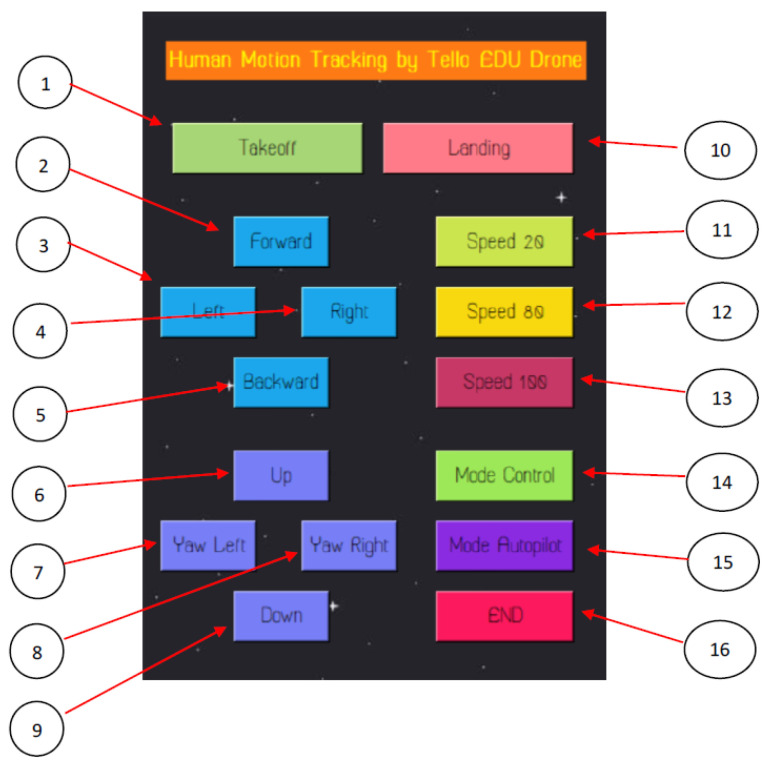
GUI design for direction, speed, manual, and automatic flight control.

**Figure 5 sensors-23-00897-f005:**
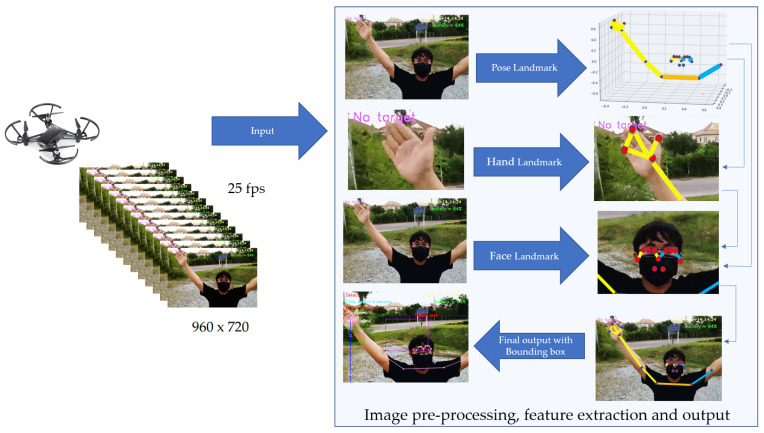
Image pre-processing, feature extraction, and output.

**Figure 6 sensors-23-00897-f006:**
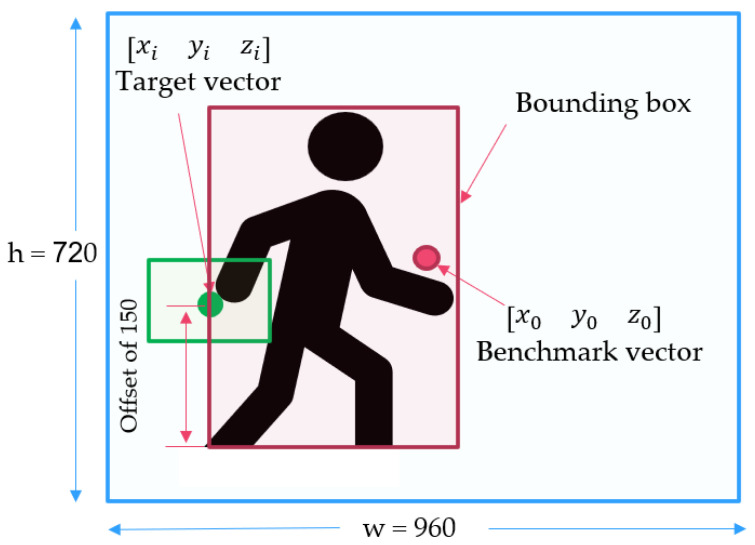
Target vector and benchmark vector.

**Figure 7 sensors-23-00897-f007:**
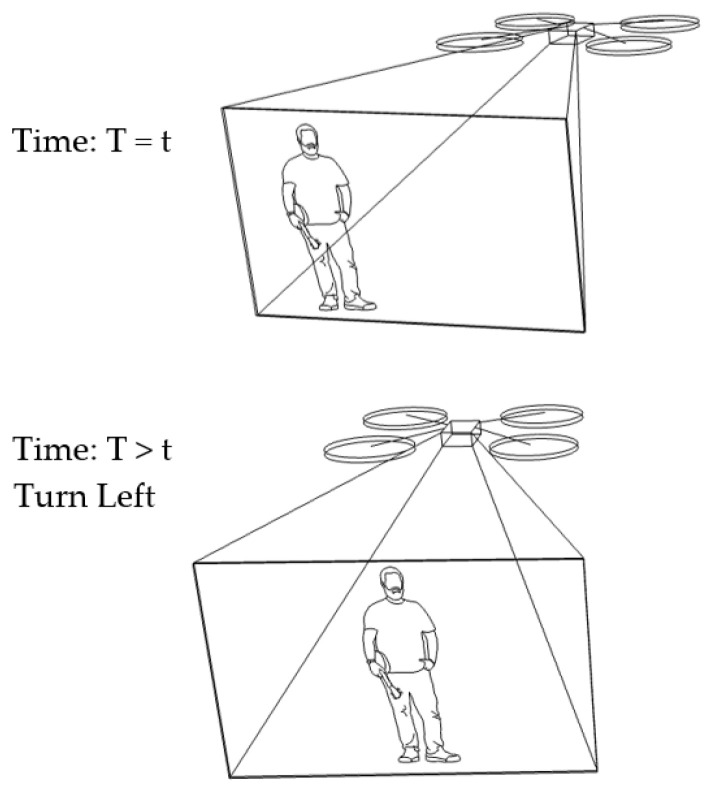
Human motion tracking when the drone turns left.

**Figure 8 sensors-23-00897-f008:**
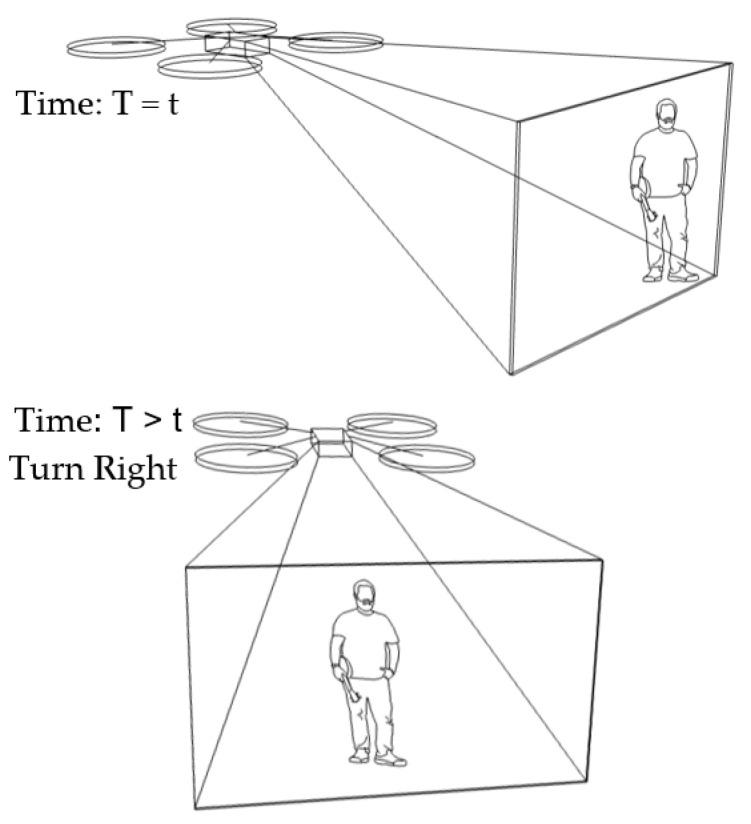
Human motion tracking when the drone turns right.

**Figure 9 sensors-23-00897-f009:**
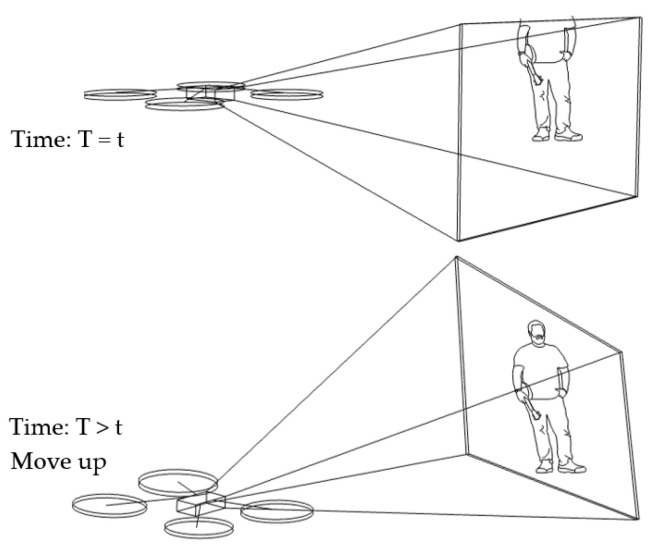
Human motion tracking when the drone moves up.

**Figure 10 sensors-23-00897-f010:**
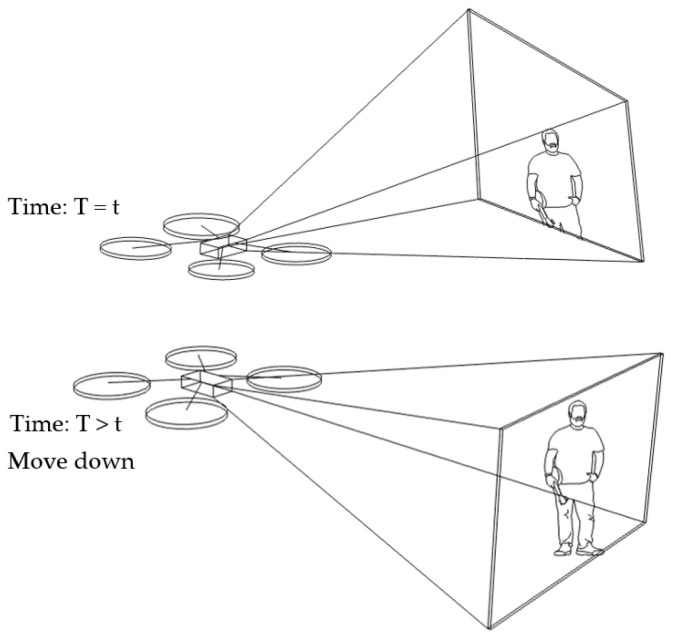
Human motion tracking when the drone moves down.

**Figure 11 sensors-23-00897-f011:**
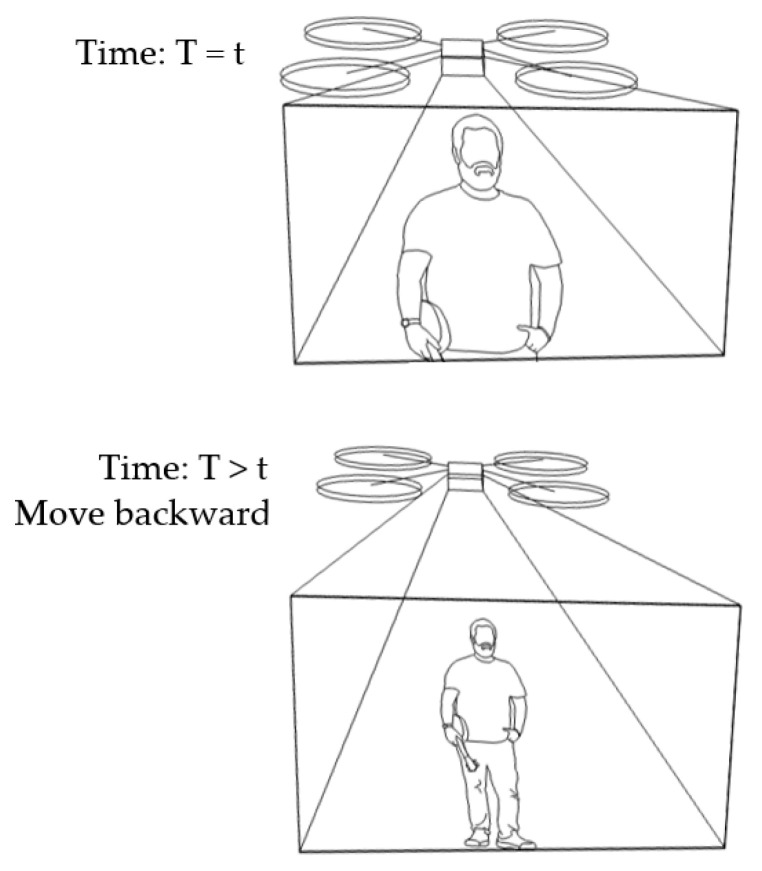
Human motion tracking when the drone moves backward.

**Figure 12 sensors-23-00897-f012:**
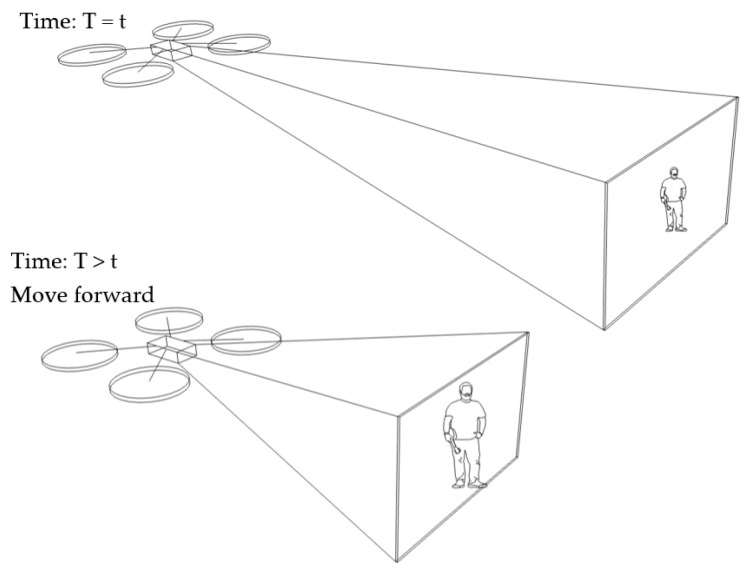
Human motion tracking when the drone moves forward.

**Figure 13 sensors-23-00897-f013:**
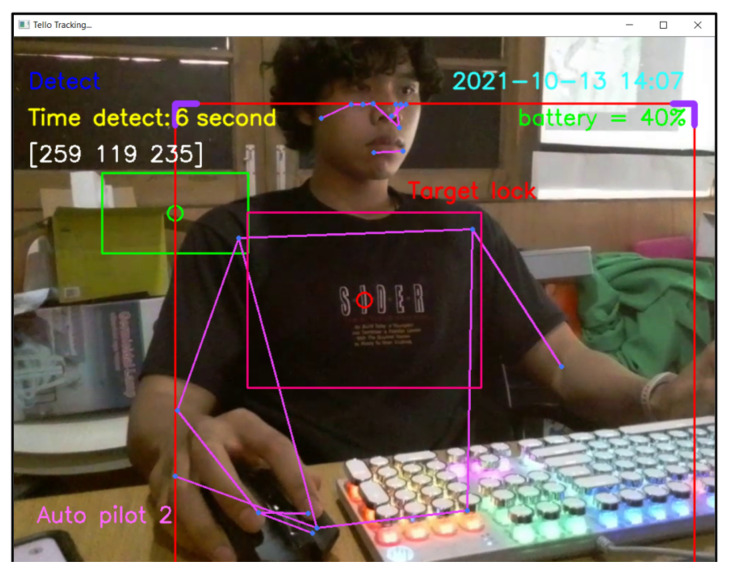
Example outputs and indicators for convenience of verification.

**Figure 14 sensors-23-00897-f014:**
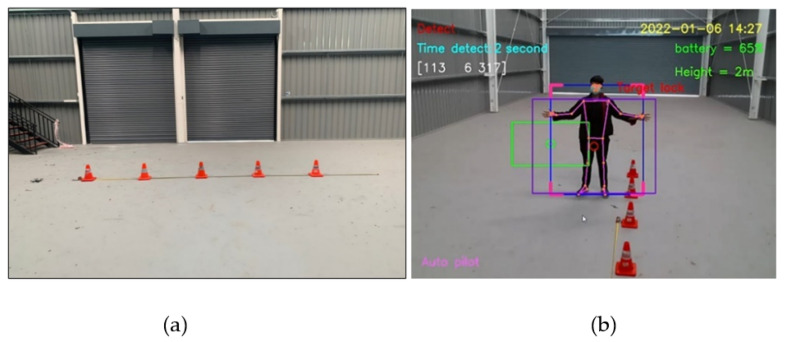
Experiments to discover proper distances between a drone and a person: (**a**) each traffic cone is equally far away (1 m separation); and (**b**) example experimental results.

**Figure 15 sensors-23-00897-f015:**
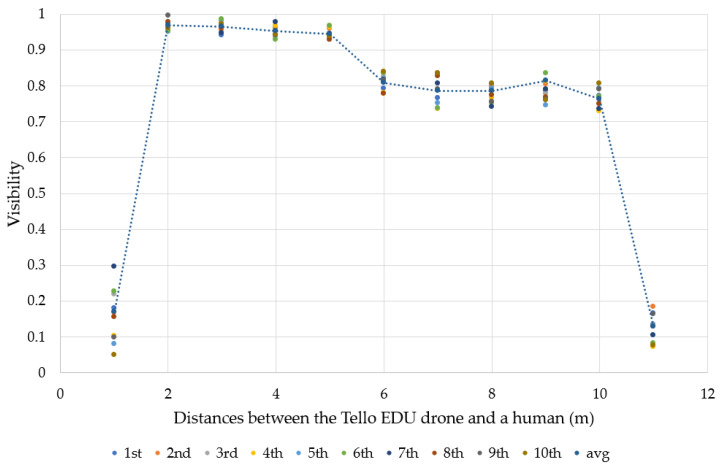
The relationship between visibility and distance (meters).

**Figure 16 sensors-23-00897-f016:**
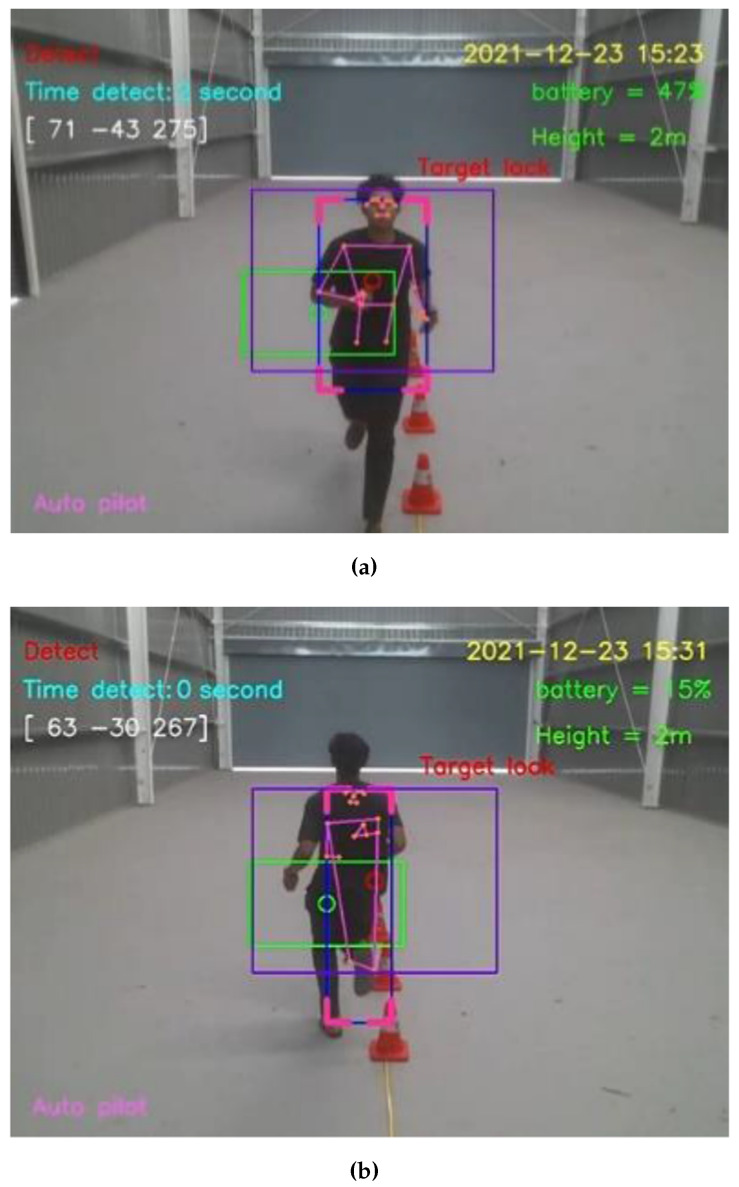
Straight moving directions: (**a**) Run forward to the Tello EDU drone (**b**) Run away from the Tello EDU drone.

**Figure 17 sensors-23-00897-f017:**
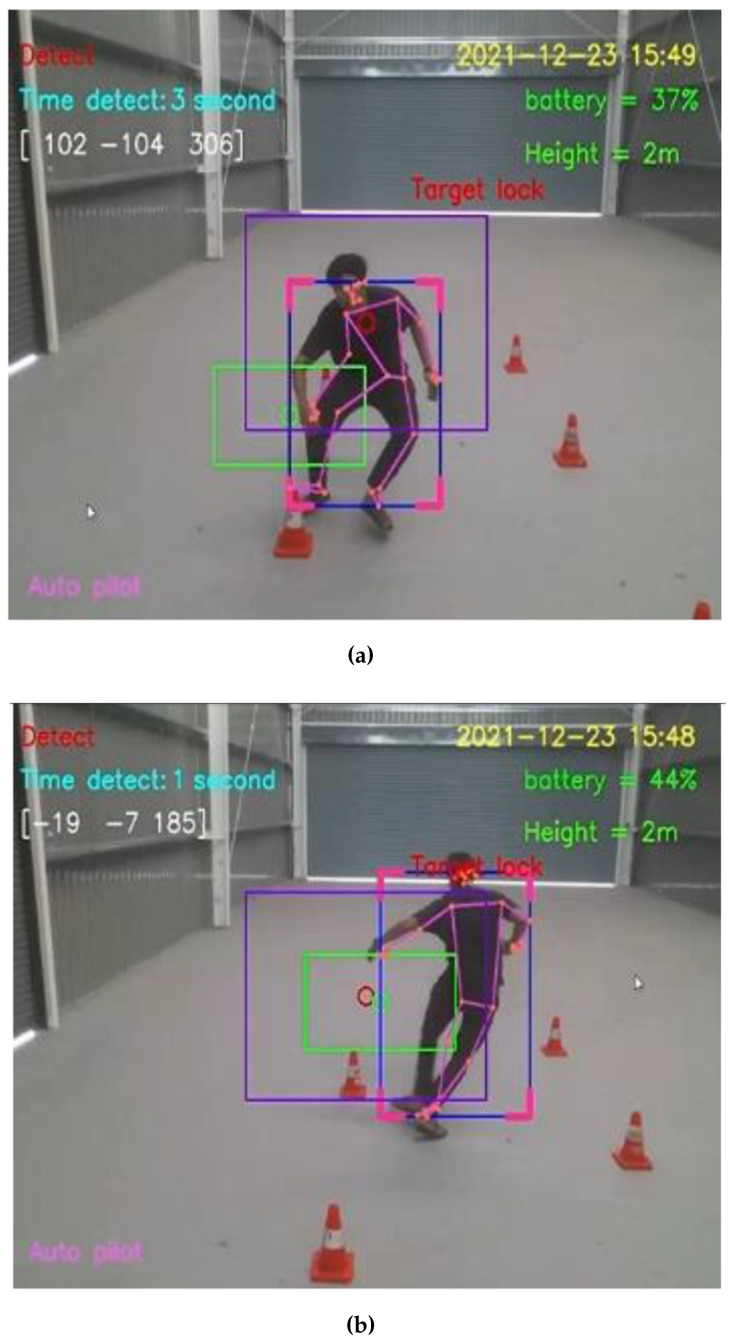
Zigzag moving directions: (**a**) Zigzag running forward to the Tello EDU drone (**b**) Zigzag running away from the Tello EDU drone.

**Figure 18 sensors-23-00897-f018:**
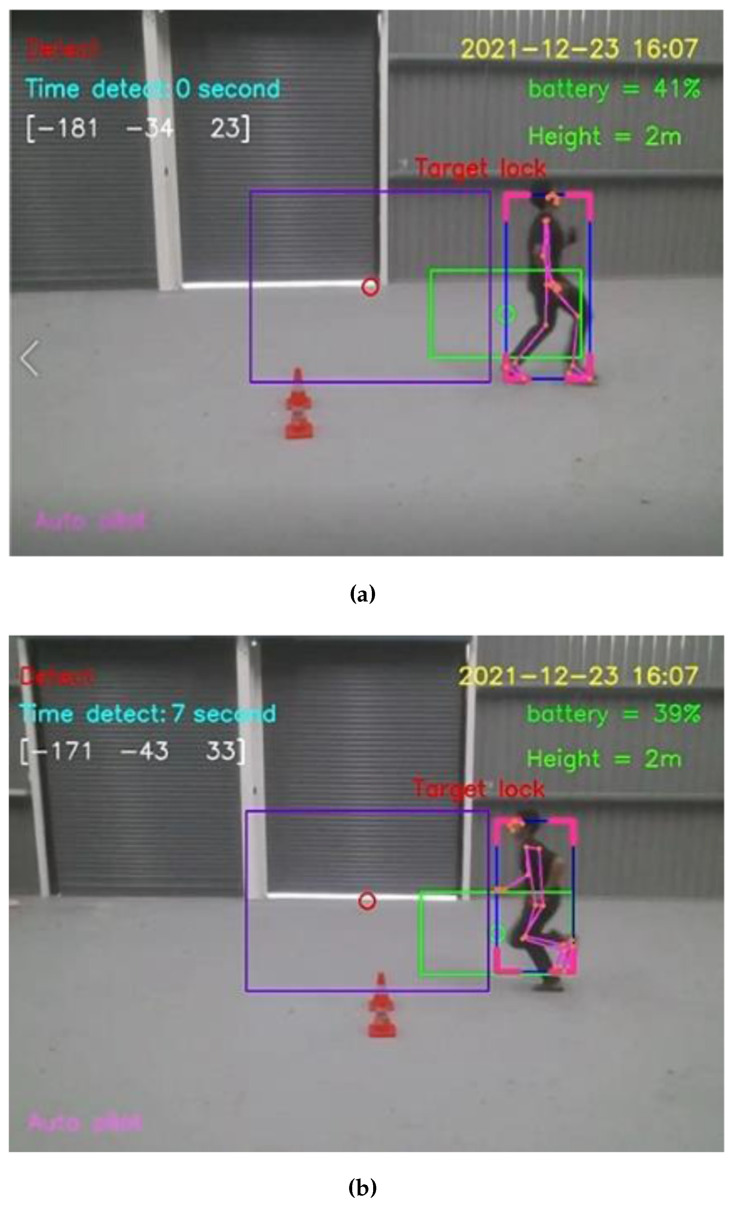
Moving through the Tello EDU drone’s camera: (**a**) Running from left to right to through the Tello EDU drone’s camera (**b**) Running from right to left through the Tello EDU drone’s camera.

**Figure 19 sensors-23-00897-f019:**
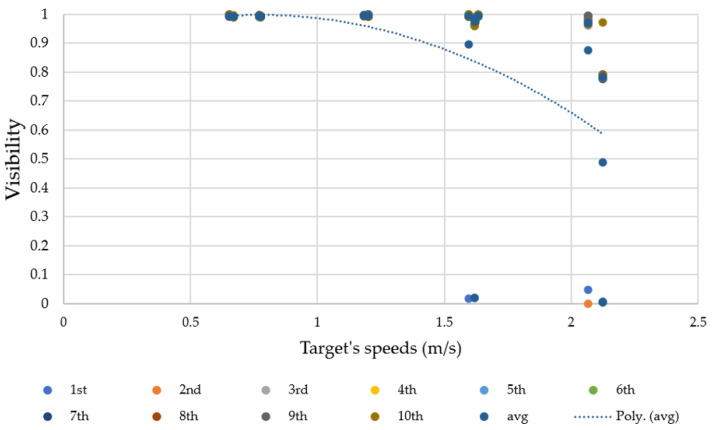
Relationship between visibility and target speed (m/s).

**Figure 20 sensors-23-00897-f020:**
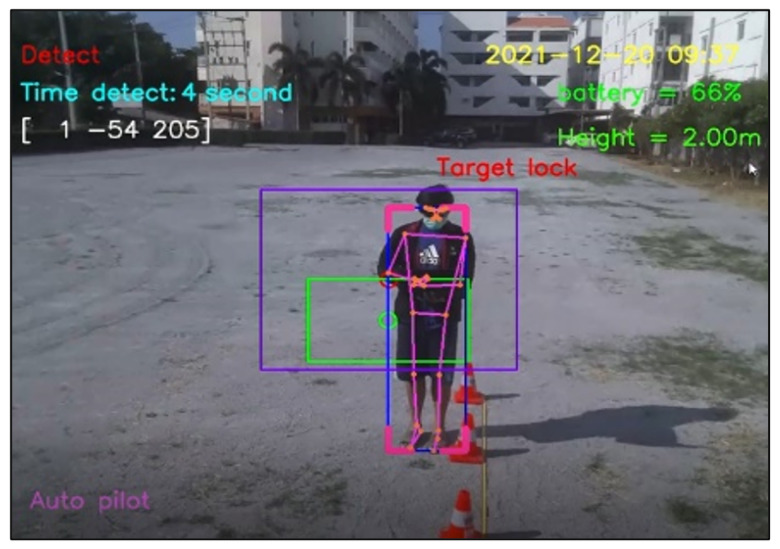
Exploration of light intensity for human motion detection in the morning.

**Figure 21 sensors-23-00897-f021:**
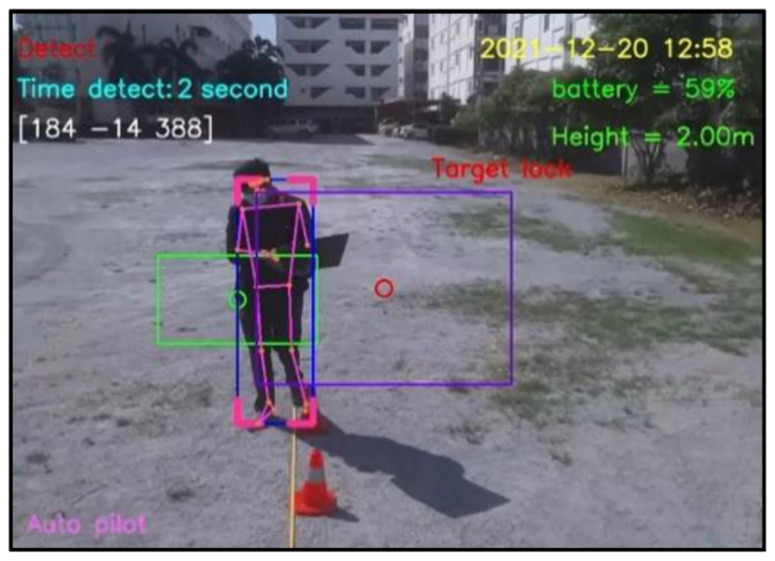
Exploration of light intensity for human motion detection at noontime.

**Figure 22 sensors-23-00897-f022:**
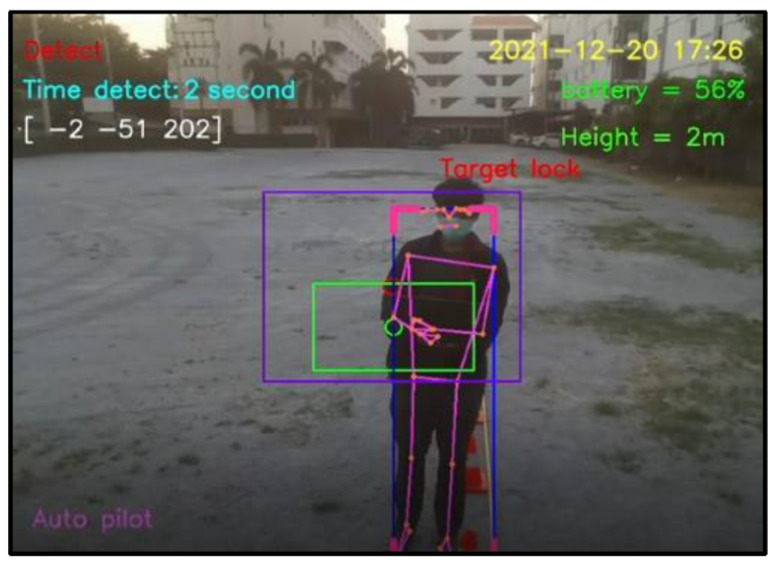
Exploration of light intensity for human motion detection in the afternoon.

**Figure 23 sensors-23-00897-f023:**
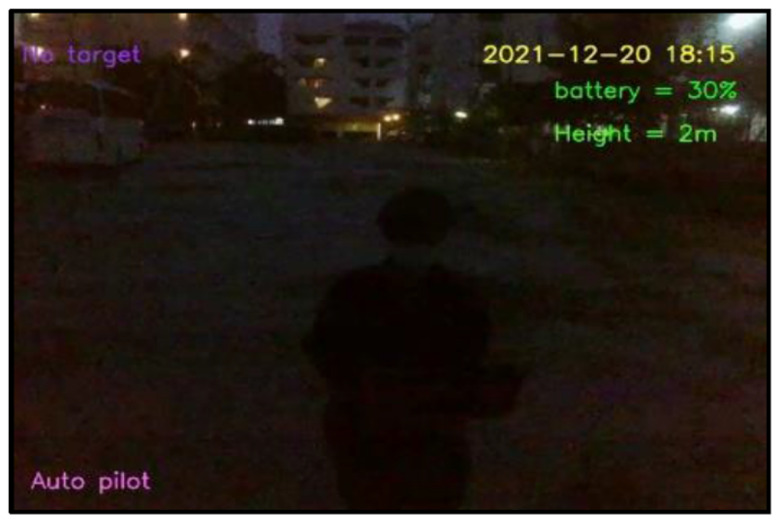
Exploration of light intensity for human motion detection in the evening.

**Figure 24 sensors-23-00897-f024:**
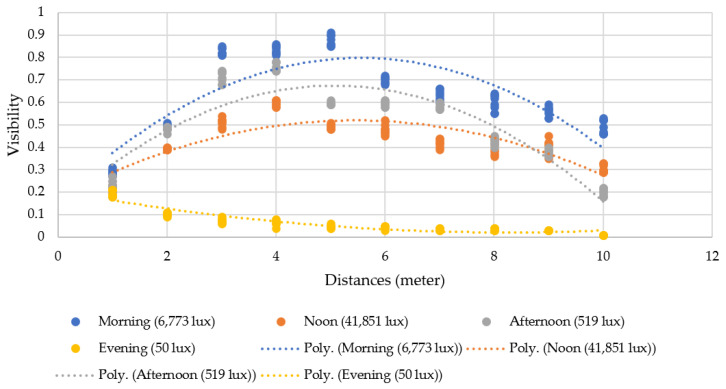
Relationship between visibility, light intensity, and distance.

**Figure 25 sensors-23-00897-f025:**
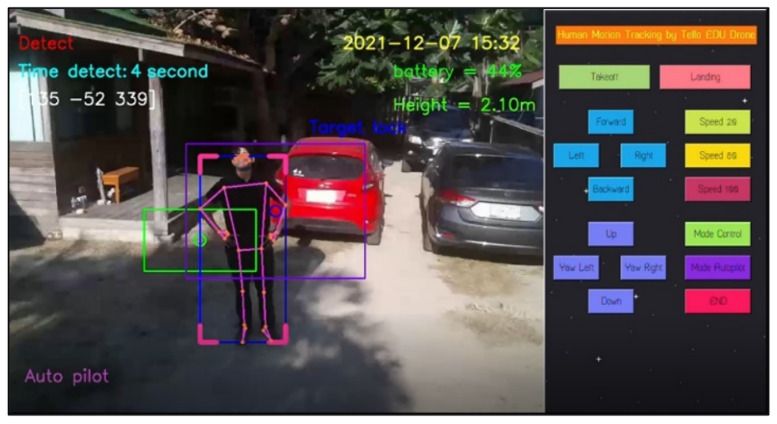
Display including control panel and first picture saved in a created folder.

**Figure 26 sensors-23-00897-f026:**
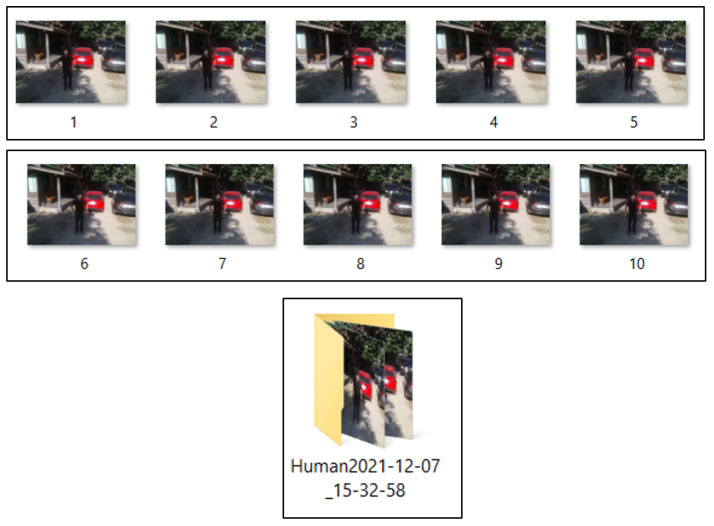
A series of 10 pictures saved in a created folder Human2021-12-07_15-32-58.

**Figure 27 sensors-23-00897-f027:**
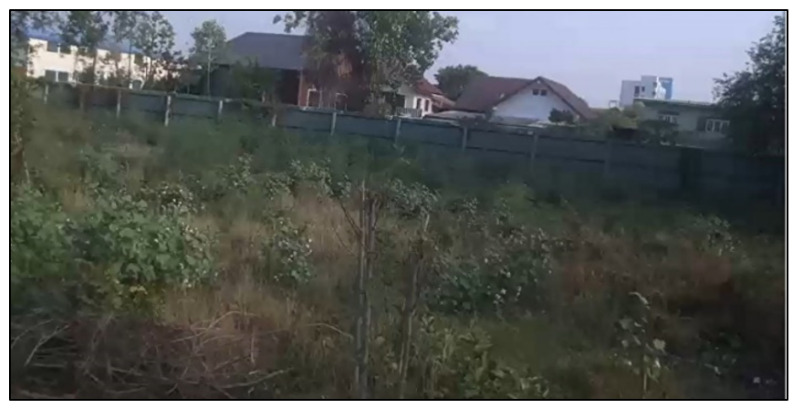
Experimental field for fault detection test.

**Figure 28 sensors-23-00897-f028:**
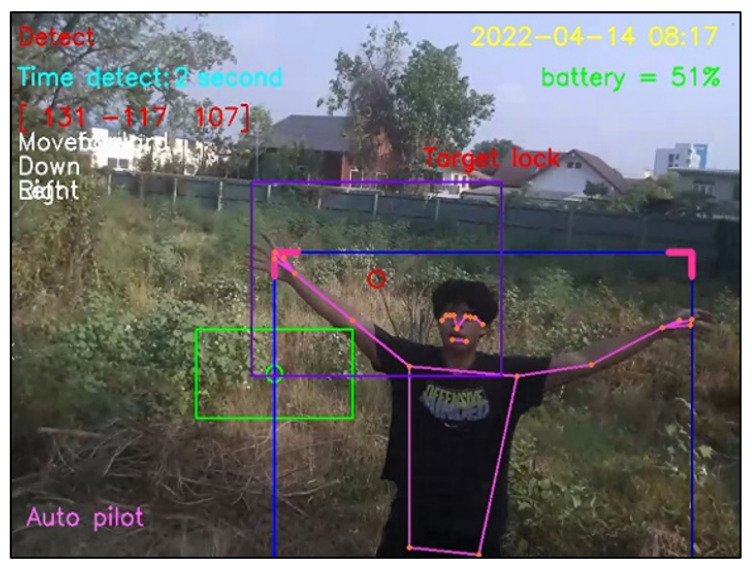
True positive detection.

**Figure 29 sensors-23-00897-f029:**
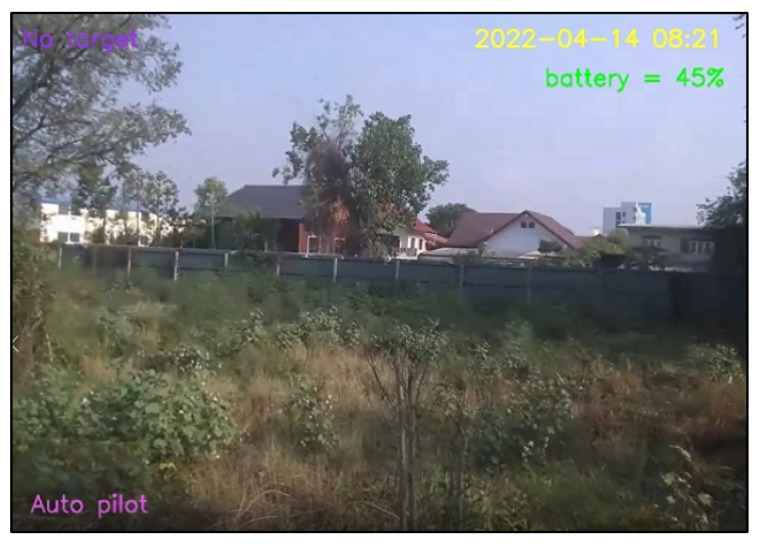
True negative detection.

**Figure 30 sensors-23-00897-f030:**
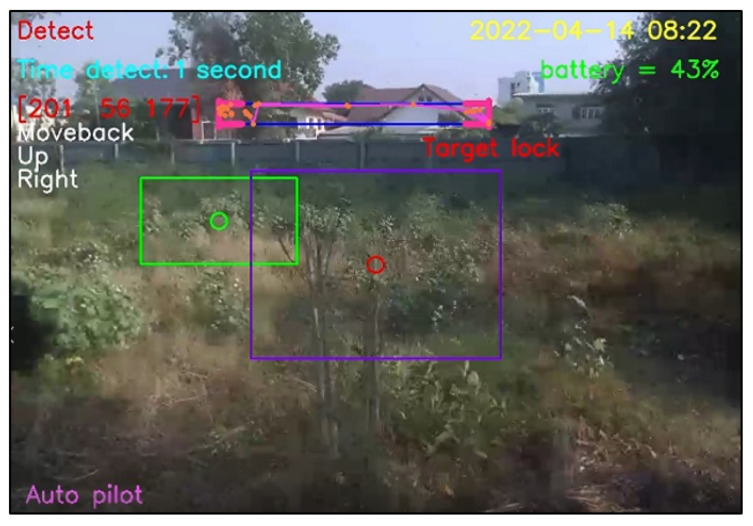
False positive detection.

**Figure 31 sensors-23-00897-f031:**
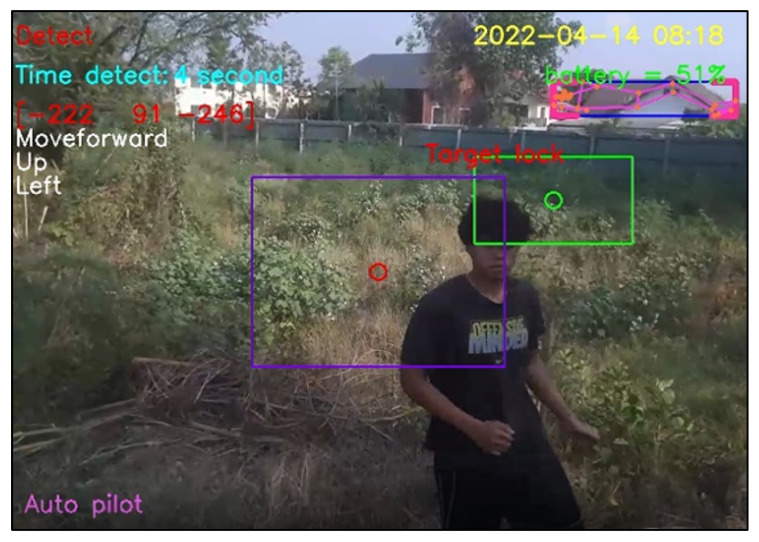
False negative detection.

**Figure 32 sensors-23-00897-f032:**
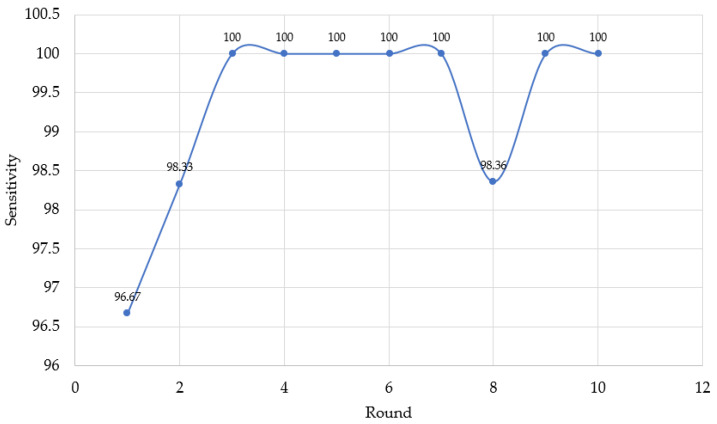
Sensitivity in human motion detection.

**Figure 33 sensors-23-00897-f033:**
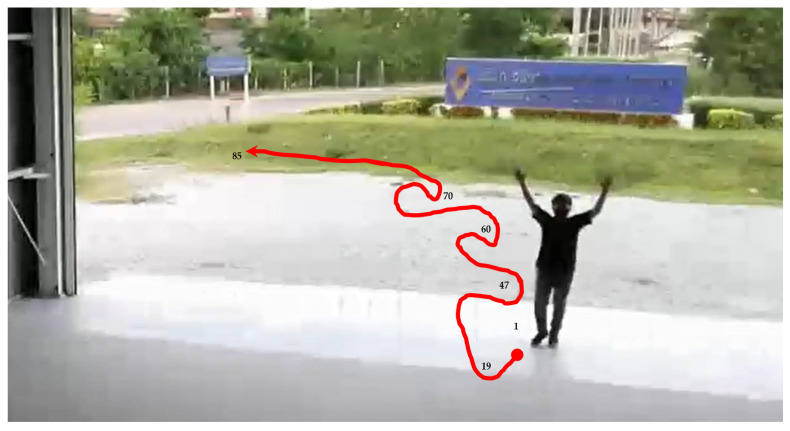
Path of automatic human motion tracking in an indoor–outdoor location.

**Figure 34 sensors-23-00897-f034:**
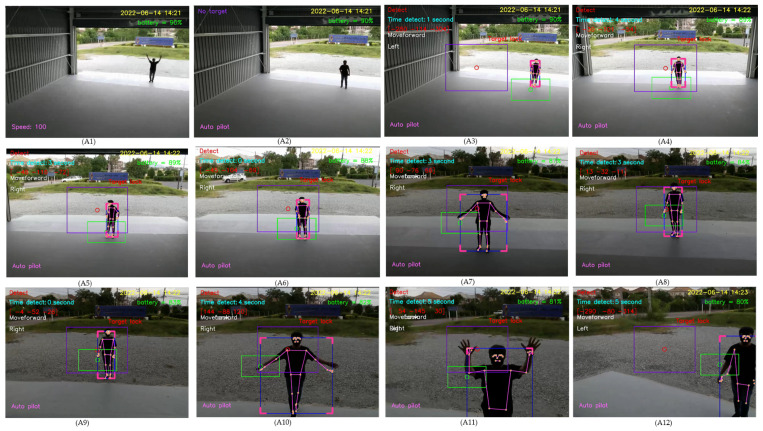
Sampling frames 1–30 of automatic tracking human motion.

**Figure 35 sensors-23-00897-f035:**
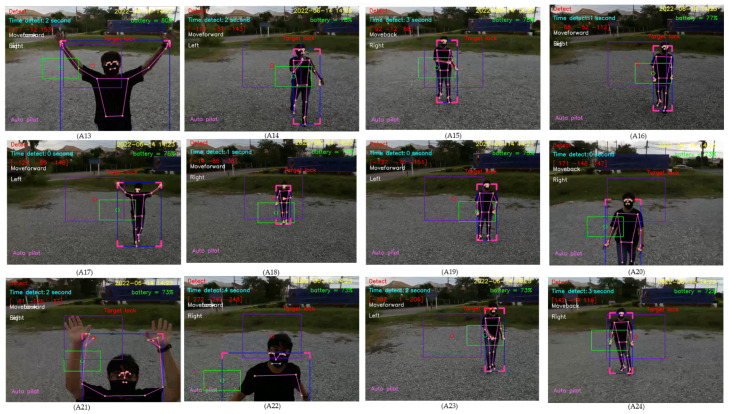
Sampling frames 31–60 of automatic tracking human motion.

**Figure 36 sensors-23-00897-f036:**
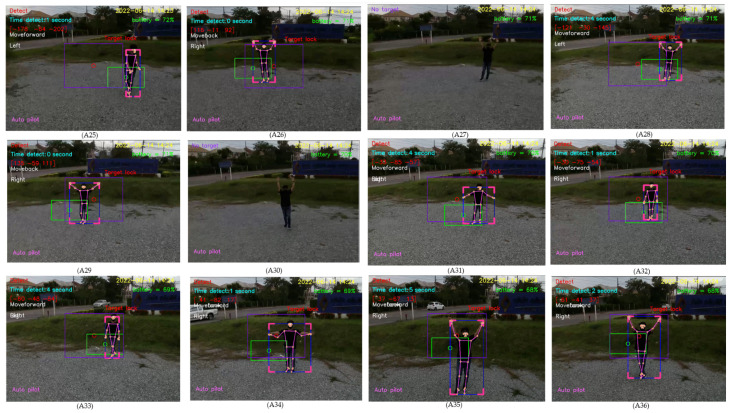
Sampling frames 61–87 of automatic tracking human motion.

**Figure 37 sensors-23-00897-f037:**
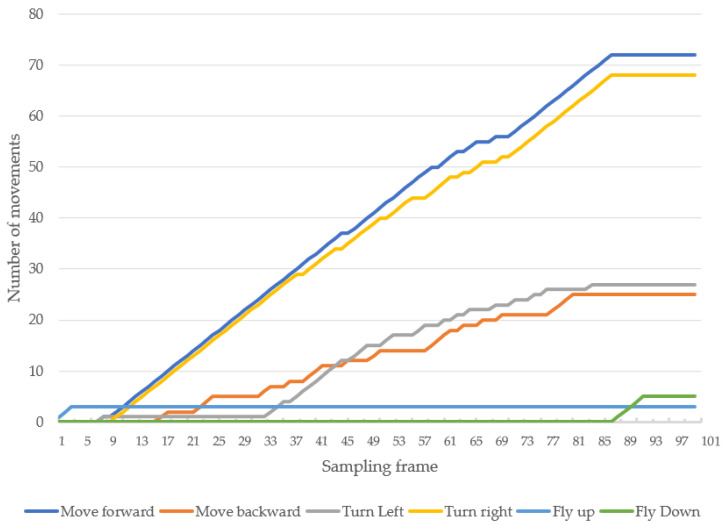
Accumulative movement numbers at various sampling frames.

**Figure 38 sensors-23-00897-f038:**
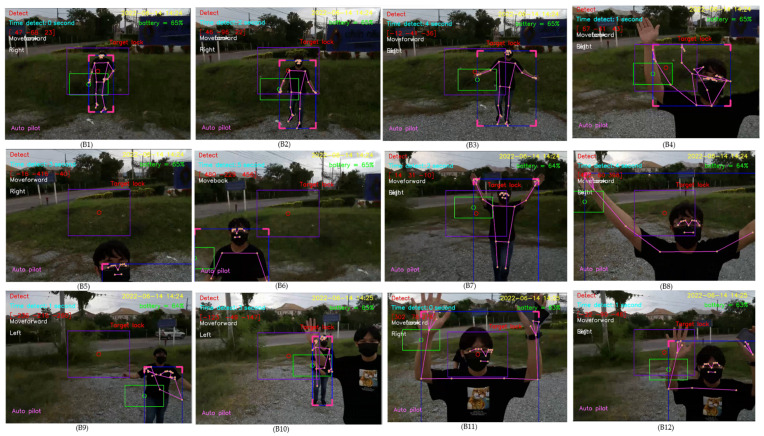
Human detection in the case when a new person comes closer than the first person.

**Figure 39 sensors-23-00897-f039:**
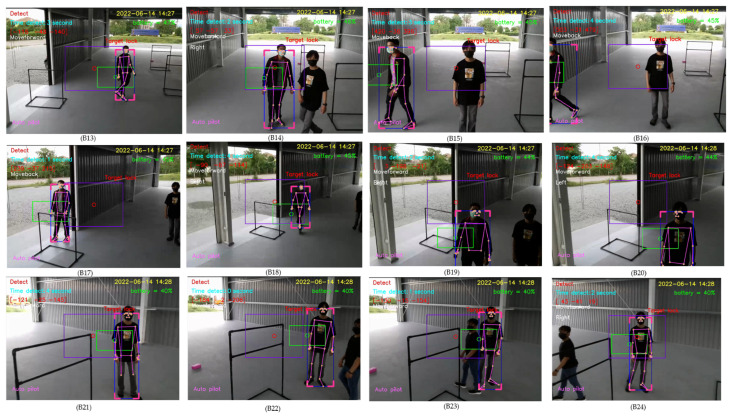
Human detection where a new person moves at the same distance as the first person.

**Table 1 sensors-23-00897-t001:** Drone movements and corresponding conditions implemented in our algorithm.

No.	Distance Conditions (Pixels)	Drone Movement
1	$\Delta x_{i}$ > 100	Turn left
2	$\Delta x_{i}$ < −100	Turn right
3	−100 < $\Delta x_{i}$ < 100	No change
4	$\Delta y_{i}$ < −55	Move up
5	$\Delta y_{i}$ > 55	Move down
6	−55 < $\Delta y_{i}$ < 55	No change
7	$\Delta z_{i}$ > 0	Move backward
8	$\Delta z_{i}$ < 0	Move forward

**Table 2 sensors-23-00897-t002:** Comparison of flight-control distances between Tello App and our design program.

Control Program	Vertical Distance No Wi-Fi	Horizontal Distance No Wi-Fi	Vertical Distance by Wi-Fi	Horizontal Distance by Wi-Fi
Tello APP on a smart phone	7.8 m	35 m	30 m	84.20 m
Our program, written in the Python language	7.3 m	33 m	13.40 m	84.30 m

**Table 3 sensors-23-00897-t003:** Comparison of human motion detection systems.

Index List	Zago, 2020 [28]	Yun, 2016 [9]	Miguel, 2017 [8]	Agrawal, 2020 [19]	Zhou, 2018 [20]	Ours
Sensitivity (%)	N/A	98.55–100	96	N/A	N/A	96.67–100
Visibility (%)	No	No	No	Yes	No	Yes
Image sensors	Webcam	Webcam	Webcam	Multi Drones	DJI Mavic Pro	Tello EDU Drone
Resolution (PX)	1920 × 1080	N/A	320 × 240	N/A	N/A	960 × 720
Required GPS	No	No	No	Yes	No	No
Frame rate (FPS)	N/A	171	7–8	N/A	24	25
CPU	N/A	Intel^®^ i7	Cortex^TM^-A7	N/A	Intel^®^ i7	Intel^®^ i7
Language	MATLAB	MATLAB	C/C++	Angular App	N/A	Python
Alert system	No	No	Yes	Yes	No	Yes
Speed (m/s)	N/A	N/A	N/A	N/A	N/A	0–2
Target distances from a cam (m)	2–6	N/A	N/A	N/A	1–10	1–10
Experiment	A room	A room	A house	A river	Variety	Variety
Intensity (lux)	N/A	N/A	N/A	N/A	N/A	519–41,851
Indoor	Yes	Yes	Yes	No	Yes	Yes
Outdoor	No	No	No	Yes	Yes	Yes
Propose	Human Fall Detection	Human Fall Detection	Elderly Fall Detection	Search and Rescue	Human Motion Capture	Human Motion Tracking

## Data Availability

Not applicable.

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
