# Peer review of "Real-Time Human Motion Tracking by Tello EDU Drone"

_sensors, 2023, doi:10.3390/s23020897_

Round 1

Reviewer 1 Report

The paper is interesting but needs improvements:

- English needs to be exstensively chacked, e.g. see lines 110-113; 115-117; 131 and following.

- lines 131-162: I think I understood what you mean but these passages need a more exhaustive explanation: 

- why 150 is your design offset?

- why for the z values a magnitude can be estimated as the same scale as a minimum x value?

- lines 288-293: need to be explained better

- lines 295-296: can you explain better?

Author Response

Dear sir,

Fisrt of all, I would like to thank you for your nice suggestion. Please see the attachment. 

Best regards,

Anuparp

Reviewer 2 Report

This paper presents a human tracking algorithm for Tell Drone robot. The results show the drone can track the human subject. It uses the size of the bounding box to keep a roughly fixed distance between the robot and the human. The algorithm is easy and straightforward. Need to point out what is the contribution of this paper. Also, it seems the tracking is largely based on detection. It will be interesting to see if the drone can keep following the same person in multi-person scenarios.

Author Response

(The authors gave the same response as above.)

Reviewer 3 Report

I would like to thank the authors for submission of this work. This seems like an interesting engineering endeavour, the manuscript is in generally well-presented and provides good depth to the methods applied and there is very good use of images and figures throughout.

My main concern with this paper is that is that the original contribution of this paper is the algorithm used to move the drone to better position so the target in the centre of the camera frame. The majority of the proposed system is more of an software engineering task in creating a Python application with the Mediapipe software. The proposed seem fairly straightforward in that there is a benchmark set of landmark positions defined as a matrix and you take the current landmarks find the difference between this and the benchmark and move accordingly based upon the rules you have defined.  

I am not totally convinced by the value of this to readers. I would be more interested in seeing how different tracking methods or threshold may provide better results i.e changing threshold in your algorthmn.

Also experiments such as visibility levels with different lighting conditions seems to have little to do with the contribution and more to do with the camera hardware and the Mediapipe Line performance, so while a nice overview of how this performs, I am not sure of the value of this to the wider community as these would likely be known a Mediapipe use with RGB cameras is well researched with the main difference being using a UAV, but other than positional angle for human detection which is not really discussed this seems a little redundant. I would be interest to see if anything else could be added to the pipeline such as image pre-processing that could help with this or a discussion on how different arial angles performs in the human detection task.

Also the paper would benefit from some tables especially in the as there are a lot of words describing the results but this makes comparison quite hard, a table would be more succinct here.

Author Response

(The authors gave the same response as above.)

Round 2

Reviewer 1 Report

The authors have improved the manuscript accordingly

Reviewer 2 Report

It is said that the contribution of the proposed human motion tracking algorithm can enhance the quality of a common drone to be a smart drone that can automatically track human motion. However, I don't see clear evidence in the paper. More experiments are needed. 

Reviewer 3 Report

I would like to thank the authors for the changes, response and re-submission of this work. The additions help to clarify certain aspects of the proposed system.

Though my main concerns remain around the significance of the contribution remain and therefore I am unsure on the interest of this to readers. For example you mention the study of visibility throughout the day in your response. Is this proving the reader with information they did not already know, i.e in that in darker conditions using RGB images from a camera performs worse than at times of the days when there is light?  Also is this specific in any way to drones and the specific camera you used as I would think this would apply for most RGB cameras that do not have some form of flash or method of lighting a scene in poor lighting, so it would be good to know how is this area of the study highlighting something that is not already well understood? 
